# A simulation-optimization framework for post-disaster allocation of mental health resources

Stephen Cunningham; Steven Schuldt; Christopher Chini; Justin Delorit

Air Force Institute of Technology, Wright-Patterson AFB, OH 45433, USA

*Correspondence to:* Stephen Cunningham (scunningham899@gmail.com)

**Abstract.** Extreme events, such as natural or human-caused disasters, cause mental health stress in affected communities. While the severity of these outcomes varies based on socioeconomic standing, age group, and degree of exposure, disaster planners can mitigate potential stress-induced mental health outcomes by assessing the capacity and scalability of early, intermediate, and long-term treatment interventions by social workers and psychologists. However, local and state authorities are typically underfunded,
understaffed, and have ongoing health and social service obligations that constrain mitigation and response activities. In this research, a resource assignment framework is developed as a coupled-state transition and linear optimization model that assists planners in optimally allocating constrained resources and satisfying mental health recovery priorities post-disaster. The resource assignment framework integrates the impact of a simulated disaster on mental health, mental health provider capacities, and the Center for Disease Control and Prevention's (CDC) Social Vulnerability Index (SVI) to identify vulnerable populations needing
additional assistance post-disaster. In this study, we optimally distribute mental health clinicians to treat the affected population based upon rulesets that simulate decision-maker priorities, such as economic and social vulnerability criteria. Finally, the resource assignment framework maps the mental health recovery of the disaster-affected populations over time, providing agencies a means to prepare for and respond to future disasters given existing resource constraints. These capabilities hold the potential to support decision-makers in minimizing long-term mental health impacts of disasters on communities through improved preparation and
response activities.

# 1 Introduction

Disaster response frameworks consist of four primary phases; mitigation, preparedness, response, and recovery, with the objective to improve disaster response capability prediction and optimal resource allocation in recovery (Zhou et al. 2018). These frameworks must also consider long-term needs for social services (such as those that target the reduction of mental health disorders as a result of the disaster itself) and long-term exposure to devastation. Unlike physical needs, which are easily identifiable and acute in the aftermath of an event, the occurrence of post-event mental health disorders can take time to manifest and can only be treated when those affected seek help. Almost all those affected by emergency situations, defined as war, natural disaster, or humanitarian crisis, experience some level of mental distress (World Health Organization, 2019). Furthermore, at any point in time, a more-acutely affected subset of this emergency-affected population (13%) experience levels of depression, anxiety, and post-traumatic stress disorder (PTSD) (World Health Organization, 2019).

Initial findings from disaster response and disaster-induced psychological stress research show mental health illness prevalence is tied to extreme-event occurrence; however, communities are consistently under-resourced to fully mitigate or respond to its effects (Benedek et al., 2007; Flanagan et al., 2011). One year after Hurricane Michael's October 2018 landfall at the panhandle of Florida, little was known regarding the mental health fallout of both victims and first responders. One-third of the affected population, in both Bay and Gulf counties, is expected to have worsening anxiety, depression, or insomnia (Rodriguez et al. 2021). In attempt to estimate the long-term effects of the hurricane, historical context can be applied. For example, 20 months after Category 4 Hurricane Irma made landfall in Florida, 17% of those in the storm's path reported being anxious while 11.3% reported signs of depression (Torres-Mendoza et al. 2021). Due to the continued prevalence of disaster-induced mental health illnesses, Torres-Mendoza et al. 2021 recommend that emergency preparedness plans emphasize mental health services especially in the context of long-term recovery.

The need for mental health service consideration in emergency preparedness is made even more evident in that preliminary data indicate that within the first two months of the start of school after landfall—December 2018—more than 700 children in the Bay County area were referred to medical services for behavioral issues (Jordan 2019). Furthermore, 70 students were taken into custody under the Baker Act, a Florida Mental Health Act designed to "reduce the occurrence, severity, duration, and disabling aspects of mental, emotional, and behavioral disorders" (The 2019 Florida Statutes). Adults seeking help after the storm experienced an array of illnesses such as anxiety, depression, and PTSD. In total, it is acknowledged that agencies and providers did not have a mental health workforce adequately sized to prevent and treat patients in the wake of Hurricane Michael (Jordan 2019).

The prevalence of post-disaster mental health illness drives the need to understand how humans respond to disaster-induced stress and what should be done to mitigate the long-term effects. The U.S. Global Change Research Program reported that first responders, children, the elderly, and those with pre-existing mental health illness are at a higher risk for weather disaster-related mental health consequences (USGCRP 2016). The report also illustrates, with strong evidence, that people who have experienced climate or weather-related disasters will develop PTSD, depression, or anxiety. These findings show the connectedness between an event and mental health disorders, and that this problem has the potential to impact anyone and in any capacity. Modeling disaster-induced psychological distress might help inform holistic response frameworks for post-natural disaster mental health recovery, which target delivering aid to those impacted by disasters with timeliness and efficiency.

This research is motivated by the potential to help communities plan for, and respond to, future disasters. Though vulnerable communities can be identified, local and state authorities are typically underfunded, understaffed, and have ongoing health and social service obligations (Flanagan et al., 2011). This limitation leads to the question: How can already constrained resources, particularly mental health clinicians, be allocated to efficiently satisfy community recovery priorities? Optimal allocations of resources will provide communities the best possible path to recovery, given available resources. This paper explores a coupled-state transition simulation and optimization model that: 1) simulates likely disaster impacts on community health and 2) optimizes the allocation of resources to address anticipated mental health clinician demand in post-disaster environments. To accomplish this, Section 2 details a literature review of the relationship between disasters and mental health illnesses. Section 3 establishes the methodology for the simulation-optimization framework, and Section 4 introduces a case study to showcase the framework's decision aid capability. Section 5 discusses the framework's implications in the broader context of optimizing disaster response, and Section 6 addresses the limitations associated with this iteration of research. Finally, Section 7 concludes with possible avenues future research may take to expand on the proposed asset assignment framework.

## 2 Literature Review

It is imperative to determine the relevant underlying factors associated with mental health and its link to disasters before constructing a model. These underlying causes that need to be explored include 1) how disasters impact an individual's mental health; 2) the link between this mental health impact and social vulnerability; 3) methods for treating those suffering from post-disaster psychological distress; and 4) the economic impact of mental health illness. The following sections discuss each of these underlying questions to motivate this research.

### 2.1 Disasters and Mental Health

A variety of factors contribute to disaster-induced psychological stress. Most prominently among these factors are an individual's proximity to the disaster and the disaster's duration and intensity (Benedek et al., 2007). The degree of psychological distress is also influenced by any physical injuries the individual may have sustained as well as the subsequent risk to their life they may have experienced (Neria et al., 2008). Individuals that experience this disaster-induced psychological stress might feel anxious, depressed, exhibit signs of Acute Stress Disorder (ASD), or have symptoms of PTSD (Mao et al., 2018). It is also possible that the individual will experience increased substance abuse and varying levels of sleeplessness, recurring intrusive thoughts, and mood changes (Simpson et al., 2011).

The relationship between disasters and mental health is further analyzed through a series of systematic reviews and meta-analyses. In a 6-year, longitudinal study of PTSD after the Indian Ocean earthquake and tsunami, the onset of PTSD was found to be one month post-disaster while the majority of those impacted recovered within three years (Arnberg, Johannesson, and Michel 2013). Interestingly, higher rates of depression and alcohol abuse were not associated with natural disaster exposure as in other studies; however, Arnberg et al., 2013 still bring attention to the persistent nature of disaster-induced negative mental health impacts.

In attempt to determine the relationship between exposed and non-disaster exposed individuals, a meta-analysis was conducted to compare psychological distress and psychiatric disorder rates post-disaster. Compared with non-exposed populations, those with exposure experienced a higher degree of psychological distress, as much as 1.84 times that of those with no exposure to a natural

disaster (Beaglehole et al. 2018). Additionally, some experience even higher rates of mental health illness than others. For example, older adults are 2.11 times more likely to experience PTSD and 1.73 times more likely to develop an adjustment disorder (Parker et al. 2016). This is consistent with other findings in the literature review that indicate the elderly are at risk in terms of disaster-induced mental health illness (Ursano et al., 2003; USGCRP 2016).

Exploring the efficacy of medical interventions to treat those affected by a disaster are imperative in rehabilitation activities. This efficacy can be used to help determine which treatments are best suited for disaster-response activities. Medical interventions ranged from community-based psychosocial programs, Neuro Emotional Technique (NET), school-based intervention, and social group work (Khan et al. 2015). One study did not show a significant improvement in mental health with the introduction of Institution-based rehabilitation therapy for earthquake survivors; however, the remaining studies did show significant improvement
in mental health outcomes due to the medical intervention (Khan et al. 2015). Specifically, Beger and Gelkopf found that 82% of probably PTSD cases improved when a school-based intervention was used to reduce stress-related symptoms of Tsunami exposure (Khan et al. 2015).

    It is possible to assign disaster-induced psychological effects into three general categories: mild, moderate, and severe distress. Mild distress causes symptoms such as difficulty in remaining asleep and elevated propensity to worry, become angry or sad.
Moderate distress causes the effects experienced in a mild case to become more extreme in the form of insomnia or anxiety. Finally, severe distress may result in cases of PTSD or major depression. As the distress becomes more severe between these three categories, it becomes increasingly important to have psychological or medical treatments available to treat those in need (Benedek et al., 2007).

With the understanding that disasters are tied to the occurrence of mental health disorders, it is also imperative to explore the likelihood of this manifestation. Prevalence of PTSD among direct victims of a disaster range from 30 to 40 percent; rescue workers, 10 to 20 percent; and the general population, 5 to 10 percent (Neria et al., 2008). It is important to note that these are averaged PTSD prevalence across three types of disasters: natural, human-made, and technological disasters. Post-natural disaster PTSD occurrence appears to be lower in human-made or technological disasters. This trend could be due to the differences in the
area of effect between the disaster types as natural disasters generally cover larger geographic areas, leading to varying degrees of impact on the affected population. Therefore, there is a stronger correlation between the level of destruction caused by the storm and the incidence of PTSD (Neria et al., 2008).

    Disaster-related PTSD also varies across population type. Apart from the distinction between rescue workers and the victims of
the disaster as Neria et al. (2008) presents, the following groups are typically more susceptible to disaster-induced psychological stress: those directly exposed to a threat of life, the injured, first responders, the bereaved, single parents, children, the elderly, women, individuals with prior PTSD, trauma, psychiatric or medical illness, and those with a lack of social support (Ursano et al., 2003). Identifying those with a lack of social support is an important consideration as it helps highlight the qualities of the environment in which the individual is living both prior to and post-disaster. These environmental qualities are additional predictors
of who may experience disaster-induced psychological stress (Bourque et al., 2006). This discussion on varying susceptibility across population types, or more generally referred to as social vulnerability, is an important consideration in determining the psychological risk factor of a disaster-affected area.

**2.2 Link to Social Vulnerability**

The Centers for Disease Control and Prevention (CDC) developed the Social Vulnerability Index (SVI) for communities to identify
at-risk populations that might need greater assistance pre-and-post disaster. The CDC defines social vulnerability as the propensity for communities to remain resilient in situations exhibiting stress on human health, while the SVI's primary use is to reduce social vulnerability by alleviating human suffering and economic loss (Centers for Disease Control and Prevention 2018).

The SVI provides vulnerability ratings at the U.S. County and Census Tract levels. Census tracts are comparable to a city
neighborhood. SVI is composed of 15 social factors across four major themes: socioeconomic status; household composition and disability; minority status and language; and housing and transportation. Vulnerability scores for each factor are aggregated into an overall SVI score. A higher score indicates a more socially vulnerable population, one that is more at risk for mental health concerns post-disaster. These tract-level ratings help disaster management organizations allocate resources to areas preparing for or recovering from either human-made or natural disasters (Centers for Disease Control and Prevention 2018).

Several studies have made use of this SVI in a variety of ways. In a post-disaster case study, the SVI was used to evaluate disaster risk-based decision making in response to Hurricane Katrina flooding (Flanagan et al., 2011). However, the SVI does not solely apply in the case of natural disasters. Studies have also investigated the relationship between heat-related illness and social vulnerability and, in a recent application, used to inform response to the novel coronavirus—COVID 19—in the State of
Washington (Lehnert et al. 2020; Amram et al. 2020). The flexibility in application of the SVI is also seen in disaster-induced mental health effects. For example, as socioeconomic status—one of SVI's four main themes—decreases, the population is more susceptible to psychological distress (Bourque et al., 2006). Social vulnerability also influences mental health resilience in a more general manner. The risk for the manifestation of disaster-induced mental health illnesses is influenced by individualistic social vulnerability (Zahran et al., 2011). Given that the CDC's SVI considers a multitude of social vulnerability indicators, the index
values can be used to help drive clinician allocation to areas of higher social vulnerability. As discussed further in Section 3, Methodology, this helps ensure clinicians are available in the areas likely to see elevated mental health illnesses post-disaster.

However, the CDC put current issues with the SVI into perspective. Though, state and local officials who plan for and respond to emergency situations have the capability to identify those in need utilizing the SVI, there are often resources constraints in terms
of both budget and personnel that limit their ability respond to an event in an optimal way (Flanagan et al., 2011). Even if these constraints are overcome, the distribution system for these resources may not be in place (Flanagan et al., 2011).

A second method of measuring social vulnerability aims to provide more context to nation-level vulnerability risk by accounting for social attributes such as socio-economic status, race, gender, age, employment, and housing considerations (Cutter and Morath
2013). Cutter and Morath 2013's Social Vulnerability Index (SoVI) uses United States Census variables, which were down selected via statistical methods designed to remove multicollinearity. Both Cutter and Flanagan et al., 2011 emphasize that social vulnerability is not derived from one demographic, but from interactions among demographic categories Additionally, both authors agree that the level of analysis must be geographically granular enough to distinguish demographical differences. The primary difference between the two methods is that the CDC's SVI can be operationalized as a toolkit, capable of informing decision-
makers with easily accessible and understandable data. While SoVI's underlying algorithms have been updated, the SVI improves based on user feedback. User feedback is an integral component of product design. The SVI becomes more useful as decision-

makers see how the toolkit adapts to their needs and provides actionable data. While the SVI helps form the foundation for the objectives of this research, it falls short in providing a method for optimizing resource allocation and considering disaster-induced mental health effects. These gaps motivate the research's overarching objective to assess risks and inform allocation decisions.

**2.3 Methods of Treatment**

Mental health care post-disaster is generally organized into three phases: early, intermediate, and long-term interventions (Hierholzer, Bellamy, and Mannix 2015). Early interventions range from Cognitive Behavioral Therapy to Psychological First Aid; intermediate interventions range from Classroom-Based Intervention to Specialized Crisis Counseling; while long-term interventions range from Cognitive Processing Therapy to Systematic Desensitization (Hierholzer, Bellamy, and Mannix 2015).
Typically, these interventions will be performed by clinicians within medical facilities and shelters, while also facilitating community-based recovery to improve upon mental health resiliency in the event of future disasters ("Disaster Behavioral Health" 2020). Cohen 2002 describes a similar approach in which treatment is distributed throughout three phases: impact, short-term, and long-term. Though different in name, these phases align similarly to those proposed by Hierholzer et al., 2015. However, Cohen 2002 introduces a new element in which treatment can target five levels of disaster-impacted individuals. Behavioral health needs
post-disaster ranges from level one, primary survivors, to fifth-level victims. Primary survivors are those who experienced the disaster first-hand while fifth-level victims are those who experience some form of distress after learning about the event (Cohen 2002). Each of the five levels will have varying recovery needs that will impact clinician allocation. While it is important to consider the three phases of treatment post-disaster, it is also imperative to introduce preventative medicine as an opportunity to decrease the mental health impact of a disaster (Math et al. 2015). Preventative medicine manifests itself in terms of readiness in
which training and equipping communities with mental health recovery tools prior to a disaster can improve the resilience of disaster-impacted individuals.

With the current understanding of disaster-induced stress and the populations vulnerable to this stress, it is also important to explore models employing post-disaster treatment options. Schoenbaum et al. (2009) explores a method of analyzing clinician-based
treatment measures that fall into the early, intermediate, and long-term interventions as described by Hierholzer et al., 2015. The study analyzed the mental health fallout from Hurricane Katrina to determine costs associated with bringing the affected population's mental health status back to a healthy level and to perform a capacity analysis of the medical support system and its availability to meet the treatment needs of the population (Schoenbaum et al., 2009).

Other possible models in addition to traditional treatment through primary health-care providers includes community-based programs and task shifting (Kakuma et al. 2011). Task shifting aims to provide some level of care to those without access to specialists. Clinicians with fewer qualifications will receive more specific training to account for the needs of the at-risk populations (Javadi et al. 2017). The case study we explore in this research focuses on providing treatment through psychologists and social workers. Given that these resource pools are limited, incorporating community-based programs and task shifting could expand the
pool of available resources to better aid the affected population in their recovery.

This paper builds on Schoenbaum et al., 2009's mental health cost recovery model and its calls for targeted resource application and advanced planning to apply these resources in an optimal way. However, the existing study does not consider optimal clinician allocation to reduce the overall economic impact of mental health illnesses.

## 2.4 Economic Impact of Mental Health Illness

Finally, the economic cost of disaster-induced mental health illness provides additional motivation for optimizing community recovery. Generally, the cost of job stress in the United States is estimated at $300 billion dollars per year, attributable to factors such as accidents, absenteeism, employee turnover, and diminished productivity (Boyd, 2011). Additionally, it is expected that individual losses are roughly $228 for each day absent from work due to poor psychological health resulting from a disaster such as a hurricane (Zahran et al., 2011). This stress cost due to both absenteeism and presenteeism is seen in a study of Major Depressive Disorder, in which monthly reduction in work and performance hours were recorded for mildly, moderately, and severely depressed workers at 37, 47.4, and 49.8 hours respectively (Birnbaum et al., 2009). Using Zahran et al. (2011) as a baseline for wage loss due to poor mental health, those with mild cases may experience a loss of $1,055 per month, while those with severe cases may lose $1,420 per month. These losses are not insignificant when considering that those affected generally live in more socially vulnerable areas and the potential enduring effects of mental health illnesses.

In addition to individualized economic loss, poor mental health can negatively impact economic growth via direct and indirect costs where direct costs include the treatment of the illness while indirect costs include income loss (Trautmann, Rehm, and Wittchen 2016). Between 2011-2030, cumulative economic output loss due to mental health illness is projected to total $16.3 trillion globally (Trautmann, Rehm, and Wittchen 2016). While this study does not provide a direct estimation for an individual's average indirect cost due to mental health, it provides support for the coupling of Birnbaum et al's estimation of hours lost and Zahran et al's wage loss estimate due to poor mental health to provide an economic loss metric that informs optimal mental health clinician allocation for wage restoration purposes). As research advances, the methodology in accounting for economic loss due to mental health illnesses can be modified through changing the inputs to the ELM, ELMod, and ELS variables. This will account for both wage changes and changes in the estimation of how many days away from work an individual will experience.

## 3 Methodology

With an understanding of the current state of the field, it is possible to develop a framework with which mental health resources can be allocated optimally in the wake of a disaster. This optimal allocation is obtained through a resource assignment framework, which is the major product of this research. A case study analysis of New Orleans, LA, explores the implementation of this framework and is discussed in detail in Section 4.

In the context of this case study, the resource assignment framework was created as a coupled-state transition simulation and multi-objective optimization model. This coupled simulation and optimization model establishes an iterative approach in simulating the mental health recovery of individuals who experienced a disaster and the subsequent optimal resource allocation given multiple decision objectives. The framework is capable of optimally allocating mental health clinicians at the census tract level, which provides enough granularity at the spatial scale for decision-makers to make coarse-grained spatial aggregations. The resource assignment framework integrates 1) simulation of disaster impact on individual mental health disorder occurrence, 2) an initial endowment of mental health clinicians and their treatment capacities, and 3) the CDC's SVI. These three pillars draw population data at the census tract level, mental health illness incidence probabilities from the National Institutes of Health, and, as previously mentioned, social vulnerability data from the CDC.

The resource assignment framework utilizes a three-phased approach (Fig. 1). Phase 1 is an event perturbance. Phase 2 is the psychological impact of event simulation, which uses the perturbance to model the population exposed to the disaster. The decision-maker, whether it be emergency planners at the national, state, or county levels, can simulate the disaster's psychological effects through probabilistic distributions of mental health illness incidence. These distributions inform a state-transition model that represents the probability that an individual who is affected by a disaster will become mildly, moderately, or severely ill. The distributions also inform the probability that the individual may recover, remain in their severity state, or change severity states with or without treatment. Once these probabilistic distributions are identified, the resulting impact on the population can be simulated to provide an estimated aggregate mental health status for the affected region. The resulting mental health status of the population derived from this simulation is then used to establish the context of the resource optimization problem. Phase 3, the resource allocation optimization, allows the decision-maker to prioritize and explore tradeoffs associated with the allocation of available mental health clinicians to treat the most severe mental health cases or to allocate these clinicians to maximize economic recovery of the disaster-affected area. Economic recovery is measured here as wage loss and includes both absenteeism and presenteeism (decrease in productivity) at work (Birnbaum et al., 2009). The preference between severity and economic loss priorities may differ based on the decision-maker, and a robust discussion of tradeoffs is provided in Section 4.

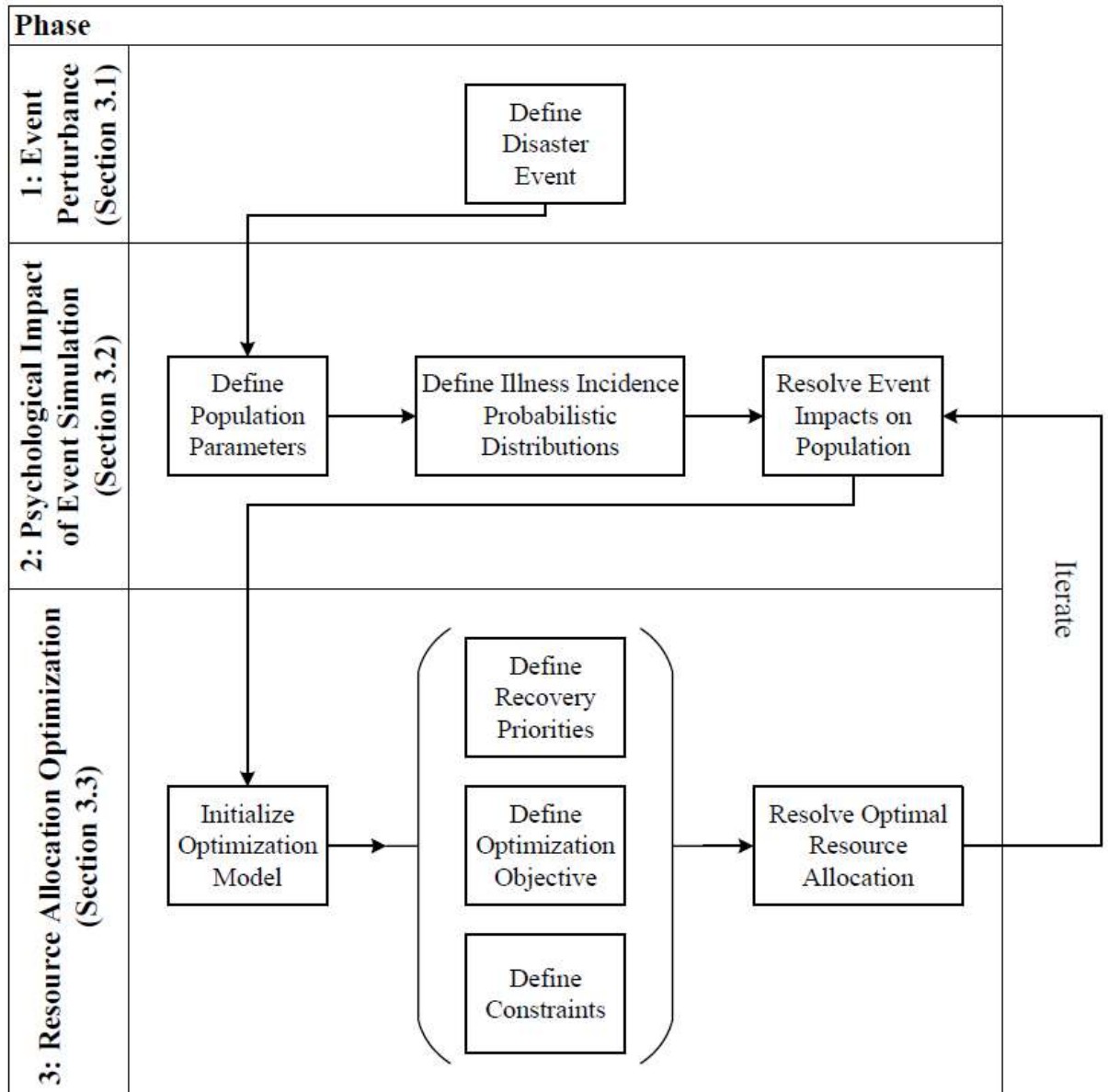

**Figure 1: The resource assignment framework provides a three-phased approach allowing iterative modeling of a community's mental health recovery post-disaster.**

The optimization produces the distribution of clinician resources at the census tract level. With this distribution of clinicians, the framework returns to the psychological impact of event simulation phase to determine the population's new mental health status after either receiving treatment based on the clinician allocation or not receiving treatment. The resource assignment framework's simulation-optimization process is designed to iterate across many time steps. This case study utilized three, six-month time steps to simulate the impact of treatment over a two-year time period consistent with Schoenbaum et al. (2009).

This proposed framework is not the first to combine simulation and optimization models in terms of resource allocation. Existing studies, most closely related to the work presented here, explore disaster resource optimization in two ways: emergency response team allocation, and disaster relief goods allocation.

In terms of emergency response team allocation, disaster support models assign and schedule response teams, such as fire, police, and ambulance teams, to aid in disaster response and recovery (Roland et al., 2010). The linear programming optimization model presented by Roland et al. (2010) aims to minimize the cost of assigning response teams to tasks. Similar studies combine simulation and linear programming optimization to simulate disaster-related infrastructure damage and the resulting optimal repair crew assignments (Brown and Vassiliou 1993). More recently, agent-based modeling has been employed to simulate how these repair crews make decisions while responding to an event (Sun and Zhanmin 2020).

Sun and Zhanmin 2020 describe that agent-based modeling can be used in the context of reinforcement learning where the agent's decision results in some reward which represents system improvement. Agent-based modeling provides an alternative approach to the simulation-optimization methodology proposed in this paper. In an agent-based modeling approach, clinicians would have the ability to choose which treatment to provide each patient, while the reward is the improvement, or degradation, of their mental health. The overall state of the system is then the quality of the community's mental health. The goal would remain the same: minimizing the mental health impact of the disaster. However, agent-based modeling would allow the clinicians to make treatment decisions and learn from the resulting mental health outcomes of those choices. This method could be increasingly useful as the complexity of the model also increases with the addition of more treatment options and treatment efficacy.

Simulation-optimization frameworks are also utilized for relief-good allocation. One framework focuses on earthquake preparedness in that it simulates the potential demand for food, water, and medicine post-disaster, then genetic algorithm to optimize the best place to store the relief goods to minimize storage and distribution costs (Ghasemi and Khalili-Damghani 2021). Alternatively, Fikar et al. (2017) looks to find the optimal location of relief good distribution centers, but also the vehicles needed to transport the goods post-disaster.

Though not directly related to disaster preparedness and response, simulation-optimization frameworks are also utilized in normal emergency room operations. Such frameworks simulate patient arrival to the emergency room and then optimize human and non-human resources alike, such as physicians, nurses, and sickbeds to best treat patients in the right capacity at the right time (Weng et al., 2011).

While the resource allocation framework proposed in Figure 1 uses a simulation-optimization approach, it is unique in both its iterative approach to re-simulate and re-optimize mental health clinician allocation based on community recovery over time as well as its ability to simulate disaster-related mental health outcomes. Additionally, and most related to Weng et al. (2011), this framework does not address cost as a decision criterion. Rather, it allocates mental health clinicians to areas of need as determined by the severity of mental health cases and the economic loss individuals may experience due to their mental health illness.

### 3.1 Event Perturbance

Phase 1 creates a disaster that informs the framework's coupled-state transition and optimization models. The event can take the form of any disaster, e.g., natural, human-made, or technological. The importance of this phase is in priming the remaining two phases with mental health illness incidence probabilities from the event. These probabilities will be discussed further in section 3.2.

There are two possible approaches in disaster identification within this phase. The first approach takes the form of a general
analysis, where the resource assignment framework utilizes uniform probabilities to generalize the selected event's impact on the
population of interest. For example, this approach could take the form of a massive event that has uniform spatial effects, similar
to the impact a large hurricane may have across a city. A second approach would be to simulate the event in a spatial context and
carry event-specific parameters forward into the state transition and optimization models. A hurricane might produce varying
damage across the city, or an explosion might cause localized catastrophic effects, which in turn, could alter the probabilistic
distributions congruent with perturbance damage. In either approach, the event characteristics, e.g., damage an illness incidence,
must have the ability to be downscaled to the census tract level. This ability allows for the framework to model events within a
spatial context. In this example, the research perturbs a uniform event for simplicity and interpretability of results.

### 3.2 Psychological Impact of Event Simulation

Phase 2 uses the disaster parameters established within Phase 1 to inform a state-transition model, which determines how the
disaster impacts the mental health of a population of interest. This model is a stochastic-dynamic simulation that determines which
members of the affected population transition from a healthy status to that of a mild, moderate, or severe status after experiencing
the event. It simulates the initial effects of an event on mental health and the transition between severity states, independent of
whether the patient has received treatment for their illness.

The first step in this phase is the definition of population parameters of interest within the area of study. Population type can be
targeted, e.g., adults, children, or first responders, or broader, e.g., a census approach where the entire population is considered
(Fig. 2). The location scope establishes the geographic boundaries of the affected population. Geographic boundaries could be at
the state, county, or census tract level. Finally, this step concludes by establishing the total population.

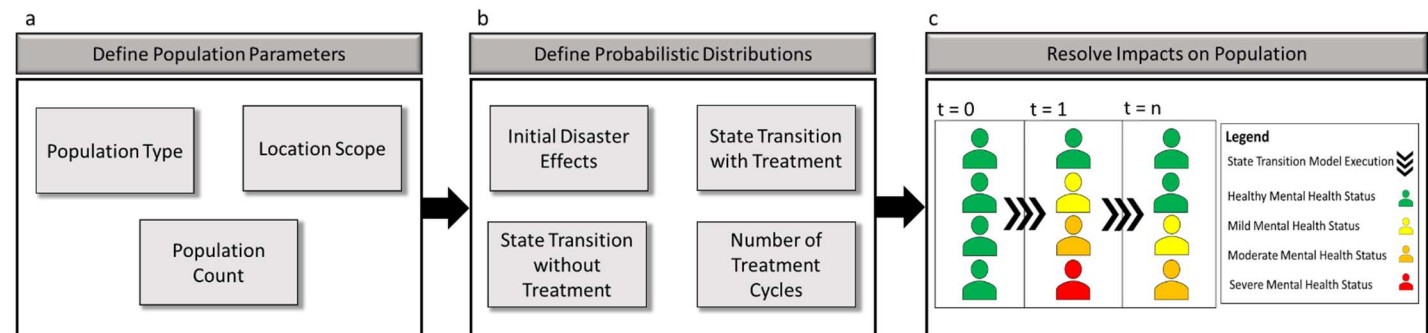


**Figure 2: Phase 2 Mental Health Illness State-Transition Model. Within C) "Resolve Impacts on Population:" t = 1: Population mental
health status pre-disaster. t = 2: Population mental health status post-disaster. t = n: Population mental health status after a period of
time in which the individuals may have received treatment.**

The second step (Fig. 2) defines the probabilistic distributions that drive the state-transition model. To begin, the initial disaster
effects are identified within Phase 1. These effects provide the state-transition model the information needed to determine which
members of the total population transition from a healthy status to a mild, moderate, or severe mental health illness. The period
under which the analysis is taking place and how many treatment cycles can be conducted within that period is also identified.
While the probability distributions can be assigned based on decision-maker preferences, the case study presented in this research
relies on three, six-month treatment cycles. It is apparent that not everyone who is impacted will seek out medical treatment. To
account for this variability, once the mental health clinicians are optimally allocated in Phase 3, the state-transition model can then

assign patients to the clinicians to account for this variability in seeking treatment. Therefore, the state-transition model will also require identification of the probability a patient will transition mental health states given that they receive or do not receive treatment.

Finally, once the population parameters and the probability distributions are set, the third step of Phase 2 resolves the impact of the disaster through stochastic-dynamic simulation. As an example, time-step 0 (Fig. 2) shows individuals who have a healthy mental health status prior to experiencing a disaster. Once the disaster occurs, they may experience a mental health state transition in time-step 1, where some individuals may remain healthy or develop a mild, moderate, or severe illness. Time-step $n$ shows another state transition potential for the affected population to recover or transition between illness states, influenced by individuals

that might or might not have received treatment. Given that a treatment cycle spans six months, the transition between one state and another occurs over this time. At the end of the treatment cycle, the decision-maker may reassess those who did not receive treatment, those who now self-identify as sick, and those who previously did receive treatment. This step is important because the onset of symptoms will vary from individual to individual. As such, those who were previously healthy for one treatment cycle may not necessarily be healthy for a future treatment cycle. Once the state-transition model is complete, optimization of mental

health resource allocation is computed.

### 3.3 Resource Allocation Optimization

#### 3.3.1 Objective

Phase 3 computes an optimal allocation of mental health resources to best treat the disaster-affected population from Phase 2, at each timestep. To accomplish this, the multi-objective model calculates optimal resource allocation tradeoffs driven by decision-

maker preference between minimizing economic loss and minimizing mental health severity.

The multi-objective resource allocation optimization model provides possible recovery opportunities that are likely to be observed at the completion of each treatment cycle, given a fixed endowment of clinicians. While true recovery is complex and individualistically specific, this basic framework provides a decision aid the field has previously lacked, which provides

suggestions of how to best allocate constrained resources for the best possible recovery opportunity either prior to, or after, a disaster has taken place.

#### 3.3.2 Model Formulation

Phase 3 formulates a multi-objective optimization model consisting of three primary steps: defining 1) decision variables, 2) objective functions, and 3) constraints. In addition to these steps, it is also necessary to discuss the inputs required to execute the

model. Figure 3 details the steps and information required to successfully execute the optimization phase of the resource management framework.

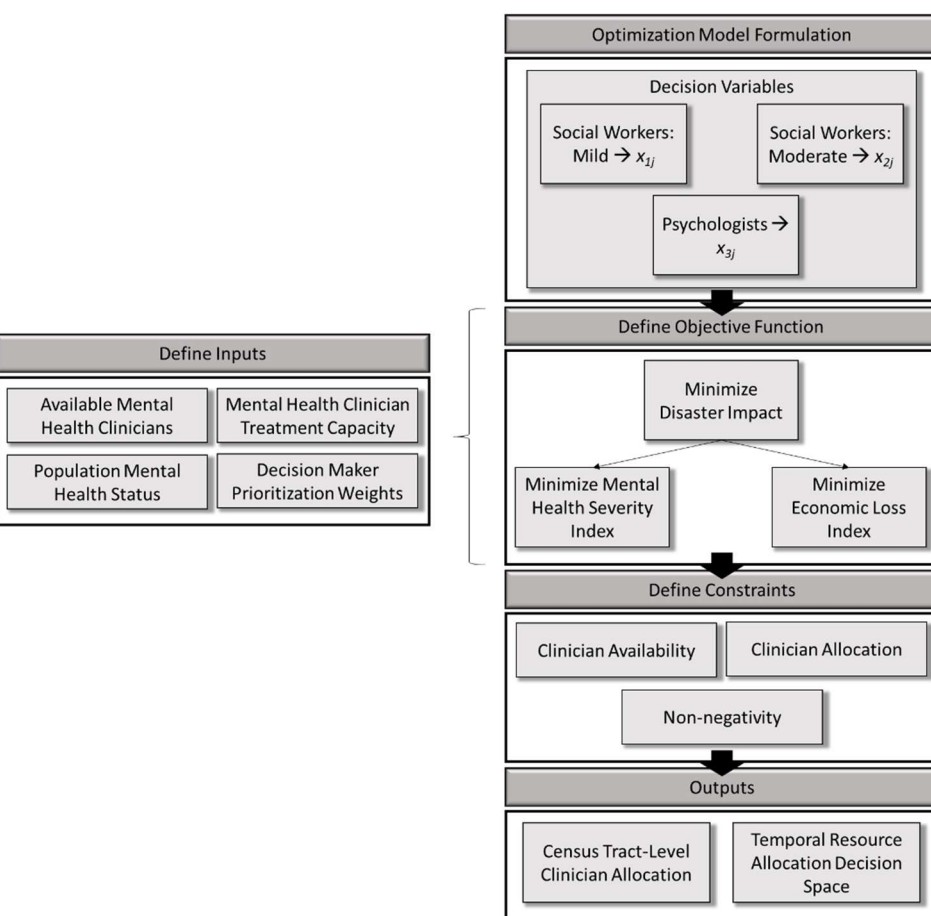

**Figure 3: Multi-objective mental health resource allocation optimization model.**

### 3.3.3 Decision Variables

First, it is important to identify the decision variables under consideration within the model. As this is a resource allocation model, the resources in question take the form of mental health clinicians. Social workers are allocated to treat mild and moderate mental health illnesses while psychologists are allocated to treat severe mental health illnesses (Schoenbaum et al., 2009). These resources are variable and take the form of the decision variable $x_{ij}$, where $x$ is the number of clinicians of type $i$, which are allocated to census tract $j$. The model requires several inputs to initialize these decision variables, which include: the type and number of mental health clinicians supplied, as well as the number of patients each clinician can treat during a treatment cycle.

### 3.3.4 Objective Function

The multi-objective optimization model utilizes integer linear programming to calculate allocation tradeoffs at the desired spatial scale, such as at the census tract level, based on decision-maker priority. Integer linear programming was chosen, as opposed to another optimization classification, to ensure a single optimal solution was achieved. Furthermore, this optimization type allows for simple setup, quick execution, and easy modification of decision criteria, which ensures accessibility for decision-makers who do not have a strong background in optimization. However, due to difficulty in determining how the relative weights between each objective will affect the allocation tradeoff, it is imperative to generate a Pareto front using varied objective weights and the subsequent set of optimal solutions (Coello 1999; Caramia and Dell'Olmo 2008).

This optimal resource allocation model satisfies two objectives: 1) minimizing the mental health impact, and 2) minimizing the economic loss of a disaster. These two objectives are measured by the Mental Health Severity Index (MHSI) and the Economic Loss Index (ELI) respectively. The MHSI is a single, global value in the model that measures the improvement in mental health status across all census tracts if clinicians are assigned to a baseline of no allocation. Alternatively, the ELI measures the economic loss of an individual who may miss work or be less productive at work due to a disaster-induced mental health illness. These indices are used to drive the allocation of clinicians to census tracts to minimize the mental health impact of the disaster, given a decision-maker's preference. This preference is operationalized through weighting criteria to provide flexibility in decision-maker prioritization towards treating for mental health severity or economic loss objectives (Eq. 1). Weight values influence the spatial allocation of clinicians, apply a zero to one scale, and must sum to one.

Minimize $MI = w_1 \times MHSI + w_2 \times ELI$ (1)

where:

$MI$ = multi-objective mental health impact of disasters

$w_1$ = MHSI objective function weight

$w_2$ = ELI objective function weight

### 3.3.5 Mental Health Severity

Mental Health Severity (MHS) is developed here as a method by which a single score can be applied to a census tract based on its number of mild ($M$), moderate ($Mod$), and severe ($S$) cases. This measure of severe equivalence quantifies the relationship between case severity and social vulnerability. To calculate severe equivalence, weight values are assigned to mild ($w_M$) and moderate cases ($w_{Mod}$), describing their relative severity when compared to a severe case. As such, $w_M$ and $w_{Mod}$ should take on values less than or equal to one. For example, a $w_M$ of 0.2 would indicate the decision-maker's valuation of a severe case as the equivalent of five mild cases. The MHSI also accounts for the social vulnerability of the census tract. A disaster's impact on mental health will be considered more severe the higher the CDC's social vulnerability index (SVI) is for a census tract. Equation 2 provides the calculation for a census tract's MHS, given that no clinicians are allocated to conduct treatment ($MHS_{NT}$).

$MHS_{NT} = SVI \times (w_M \times M + w_{Mod} \times Mod + S)$ (2)

Equation 3 provides a measure of how MHS improves with clinician allocation ($MHS_T$). Equation 4 calculates Unmet Demand (UD), which is the latent demand for treatment within census tracts after clinicians have been allocated. Please note that all variable descriptions are found in Table 2.

$MHS_T = SVI \times UD$ (3)

$UD = (S - P_{Cap} \times P) + w_M \times (M - SW_{Cap} \times SW_M) + w_{Mod} \times (Mod - SW_{Cap} \times SW_{Mod})$ (4)

where:

$P$ = Number of psychologists allocated

$P_{Cap}$ = Number of patients a psychologist can treat

$SW_M$ = Number of social workers allocated to treat mild cases

$SW_{Mod}$ = Number of social workers allocated to treat moderate cases

$SW_{Cap}$ = Number of patients a social worker can treat

Finally, MHSI is calculated in Equation 5. This becomes an index [0,1], where a value of zero represents a complete reduction of MHS across all census tracts (J), and a value of one indicates no improvement.

$$MHSI = \frac{\sum_{j=1}^{J} MHS_{T,j}}{\sum_{j=1}^{J} MHS_{NT,j}} \tag{5}$$

### 3.3.6 Economic Loss

Next, the ELI measures the improvement in economic loss, which is defined by both wage loss of an employee who may miss work due to mental health illness, and economic loss borne by the employer, due to reduced worker productivity (Birnbaum et al., 2009). Equation 6 shows the Economic Loss (EL) of a census tract, where EL is the economic loss ($) due to expected productivity days lost multiplied by the mean daily income of the census tract.

$$EL_{NT} = EL_M \times M + EL_{Mod} \times Mod + EL_S \times S \tag{6}$$

where:

$EL_M$ = Economic Loss of mild cases measured in daily productivity loss ($)

$EL_{Mod}$ = Economic Loss of moderate cases measured in daily productivity loss ($)

$EL_S$ = Economic Loss of severe cases measured in daily productivity loss ($)

Equation 7 provides a measure for how EL improves with clinician allocation, which is similar in concept to UD in that it determines the total EL of a census tract when considering the individuals who have not been treated by a mental health clinician.

$$EL_T = EL_S \times \left(S - P_{Cap} \times P\right) + EL_M \times \left(M - SW_{Cap} \times SW_M\right) + EL_{Mod} \times (Mod - SW_{Cap} \times SW_{Mod}) \tag{7}$$

Like MHSI, ELI ranges from zero to one where a value of one indicates the absence of effective treatment across all census tracts, resulting in full economic loss. Equation 8 details the final ELI calculation.

$$ELI = \frac{\sum_{j=1}^{J} EL_{T,j}}{\sum_{j=1}^{J} EL_{NT,j}} \tag{8}$$

### 3.3.7 Model Constraints

Once the optimization model objective functions are defined, the model's constraints must be established. Again, the framework holds the flexibility to add and remove constraints to tailor the optimization to the needs of the decision-maker. However, this iteration baselines three constraints: 1) clinician availability, which prevents the number of clinicians allocated from exceeding the number available to allocate; 2) clinician allocation, which prevents the optimization model from assigning more clinicians than there is demand within each census tract; and 3) non-negativity, which prevents any decision variable from holding a value less than zero. Though these three constraints allow the model to achieve an optimal solution, other constraints may be added. For example, a constraint could be written to ensure a minimum number of clinicians are supplied to each census tract.

## 4 Case Study

### 4.1 Case Study Introduction

As proof of concept, this research utilized the resource assignment framework in a simulated case study. The disaster analyzed was a hurricane that impacted Orleans Parish, Louisiana, where all measures were taken at the census tract level. The 2016 CDC SVI data for Orleans Parish, Louisiana were used to inform the model in terms of social vulnerability scores [min: 0; max: 1], population size [0; 7,381] (Fig. 4), and mean income [$3,710; $111,631] at the census tract level.

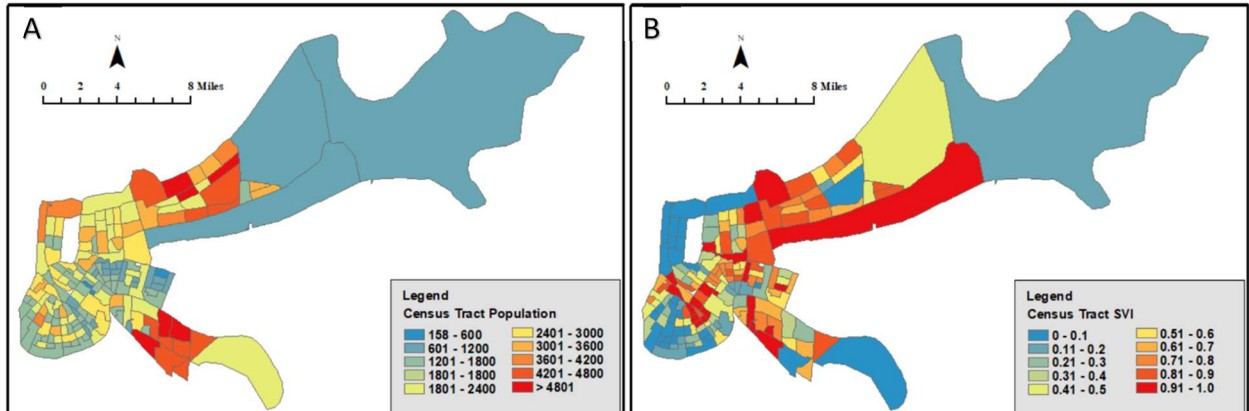

**Figure 4: A) New Orleans population by census tract. B) New Orleans SVI scores by census tract.**

In 2016, Orleans Parish consisted of 177 census tracts; however, this research considered only 172 tracts due to missing population data (3) and missing SVI data (2). Transition probabilities for expected rates of incidence and recovery were applied uniformly in the stochastic-dynamic simulation to resolve the psychological impact of a disaster on the population of interest (Schoenbaum et al., 2009). Table 1 provides an example of mental health illness incidence that could be expected 6-12 months post-hurricane. Please refer to Schoenbaum et al. (2009) for the complete table of probabilities used in this case study.

**Table 1: Mental Health Illness Incidence Probability Sample (Schoenbaum et al., 2009)**

| Time Post-Disaster | Illness Severity | Incidence Probability (Age 20+) | Incidence Probability (Age 5-19) |
|---|---|---|---|
| 6 – 12 Months | Mild/Moderate | 25% | 30% |
| 6 – 12 Months | Severe | 5% | 10% |

The case study utilized social workers and psychologists as the resources it allocates to treat mental health illnesses. The number of mental health clinicians available to treat patients after a disaster can vary; however, for the purposes of this research, clinician availability was determined by the number of clinicians registered within the state of Louisiana as of 15 Oct 2020 [891 social workers, 52 psychologists], which is given by the National Practitioner Data Bank from the U.S. Department of Health & Human Services (Singh 2020). Each social worker will have a treatment capacity of 18 patients (Whitaker et al. 2004). Psychologist capacity was set to 20 cases to avoid burnout (Kok et al. 2015).

The first treatment cycle begins six months after the hurricane, and recovery projections are provided at six-month intervals to 30 months post-hurricane. As such, the first treatment cycle begins six months after the hurricane, as a reflection of the time it takes

for mental health effects to manifest within each individual. However, this does not mean treatment must wait until six months have passed, this is one of the many variations the resource assignment framework is capable of handling. Finally, severe equivalence weight values include 0.2 for mild cases ($w_M$) and 0.7 for moderate cases ($w_{Mod}$). These weights can be any value between zero and one, depending on decision-maker preference and data availability.

490

This case study varies the objective function weight values for each treatment cycle to show how potential recovery might change, given varying decision-maker priorities over the two-year period. However, the number of available clinicians and probability distributions remained constant to keep complexity low and limit variability. Both elements can be varied as desired or necessary. Similarly, variability can be introduced in economic loss by sampling from a distribution of incomes earned at the census-tract level. In this proof of concept, the mean income of each census tract was used.

495

## 4.2 Model Evaluation

The coupled model produces pairings of mental health severity and economic loss outcomes for many time steps, and it illustrates how decision-maker preference variation impacts recovery. Figure 5 illustrates optimal recovery tradeoffs between the optimization objectives MHSI and ELI, 12 months post-hurricane. Using 11 weight combinations [0%, 100%; 10%, 90%; 20%, 80%; etc.], the resource assignment framework computed 11 optimal solutions, each with a unique impact on mental health and economic recovery. The number of optimal solutions varies based on the decision-maker-defined weight increments between MHSI and ELI. Points B, C, and D along the Pareto front illustrate the tradeoffs between varying preferences. Focusing on the extremes, full preference for mental health severity (B) provides the maximum possible recovery, measured in the severity of cases, while minimally improving Orleans Parish's economic loss. Alternatively, devoting full preference to economic loss recovery (D), Orleans Parish could maximize economic recovery with a $74.13 million improvement over point B; however, this causes Orleans Parish to experience 9,661 more severe equivalents than point B. Point C, which represents a case of equal preference between MHSI and ELI, could allow Orleans Parish to see a $54.47 million and 8,164 severe equivalent improvement from their status six months earlier. Varying preference from the equal weighting at point C yields a greater change in both objectives with diminishing returns in approaching the extremes.

510

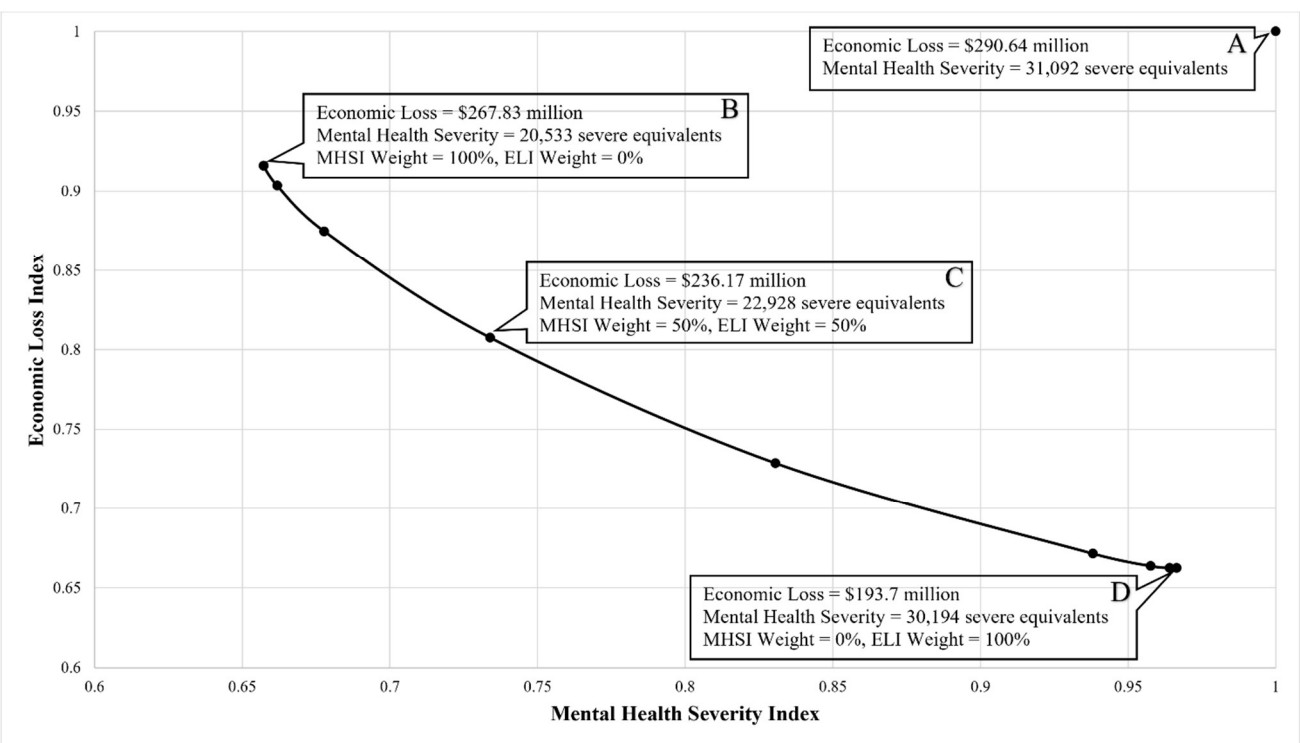

**Figure 5: Possible measures of recovery for mental health clinician allocation in Orleans Parish, LA, at 12 months post-hurricane. Each point shows a 0.1 shift in preference. A) Economic Loss and Mental Health Severity at the 6-month, post-hurricane resource allocation decision point. B) New Economic Loss and Mental Health Severity at 12 months if Mental Health Severity is given 100% preference. C) New Economic Loss and Mental Health Severity at 12 months if 50% preference is given to each priority. D) New Economic Loss and Mental Health Severity at 12 months if Economic Loss is given 100% preference.**

The temporal mental health resource allocation decision space provides decision-makers with sets of Pareto fronts at the beginning of each round of treatment, and it may be thought of as a long-term recovery model (Fig. 6). This decision space includes all possible outcomes for the range of preferences the decision-maker may be able to take throughout the 2-year recovery period. Given that Orleans Parish could experience $290.64 million in economic loss due to worker absenteeism and presenteeism 6 months post-hurricane, the best possible economic recovery is by $238.34 million to a loss of $52.296 million at point (F). Alternatively, the best possible mental health severity recovery is by 22,907.8 severe equivalents to a remainder of 8,184.2 severe equivalents at point (E). This severe equivalence equates to roughly 16,500 mild cases, 12,900 moderate cases, and 20,900 severe cases out of a total population of 381,002.

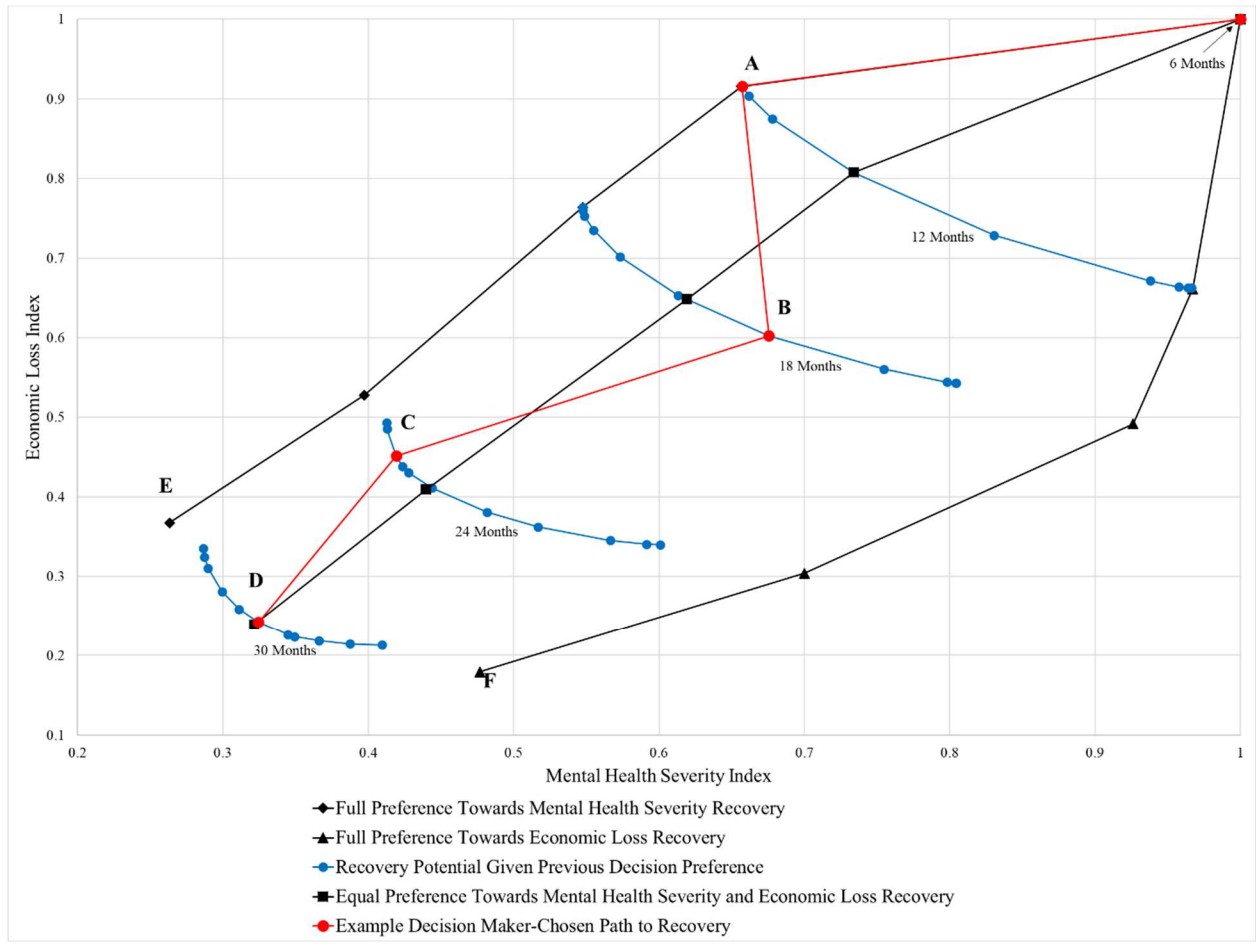

**Figure 6: Temporal mental health resource allocation decision space. Pareto fronts show potential recovery in months past the hurricane. A) Full preference for Mental Health Severity. B) 60% preference towards Economic Loss. C) 80% preference towards Mental Health Severity. D) Equal preference for Economic Loss and Mental Health Severity. E) Economic Loss: $106.81 million per six-months; Mental Health Severity: 8,184.2 severe equivalents. F) Economic Loss: $52.296 million per six-months; Mental Health Severity: 14,892 severe equivalents.**

### 4.3 Varied Recovery Paths

It is also possible that a decision-maker will want to select a path with varied preferences at each treatment round, to address the needs of their community over the two-year period. The red line in Figure 6 shows a theoretical path a decision-maker could take to achieve recovery. The subsequent blue Pareto fronts represent the potential recovery at that time step given the clinician allocation decision made at the previous time step.

In this scenario, the decision-maker understands that by 6 months post-event, the community has been impacted such that action must be taken to reduce economic losses and case prevalence. In an attempt to reduce the severity of mental health illness across the parish, the decision-maker chooses to fully prefer mental health recovery over economic loss in the first allocation (path along the red line from 6 months to A, 12 months post-hurricane). For an example of the resulting simulation-optimization output for the allocation decision made at point A, please reference Supplement A. Once this treatment cycle concludes at 12 months post-hurricane, the decision-maker is encouraged by the status of the parish's recovery. As such, the decision is made to weigh economic recovery more heavily while still ensuring the severity in cases retains some consideration in the next treatment cycle (B, 18 months post-hurricane). At this point, the decision-maker realizes that despite the gains in economic recovery, Orleans Parish has regressed

slightly in mental health severity. The decision-maker understands that this could be due to delayed illness incidence from those who were previously healthy at month 12. In an attempt to rectify this regression, the decision-maker redeploys the clinicians in favor of greater mental health severity recovery (C, 24 months post-hurricane). The decision-maker now sees that Orleans Parish is trending towards full recovery, with improvements realized in both ELI and MHSI. To ensure the community continues on this path, the decision-maker makes a final choice to give equal preference toward each objective (D, 30 months post-hurricane). At 30 months post-hurricane, the decision-maker can expect Orleans Parish to reduce economic losses by $220.22 million per-six months and have fully treated 21,005 severe equivalents.

The path this decision-maker took through the recovery decision space illustrated two important points: 1) the decision-maker's selected course of action may not always improve recovery and 2) the possible outcomes narrow as more allocation decisions are made, which makes outcomes towards the extremes impossible to achieve under a static resource endowment constraint. Fortunately, the decision-maker does not need to make these decisions blindly. Using the resource assignment framework, the decision-maker can simulate the possible consequences of actions taken.

Alternatively, the decision-maker can also simulate the possible consequences of not acting. A do-nothing approach, where the population is left to recover on its own, is relevant to determine the comparative value of the resource assignment framework. Orleans Parish could expect to avoid $35 million in a combination of both work absenteeism and presenteeism in the first 24 months post-hurricane if the endowment of mental health clinicians and equal MHSI and ELI preferences presented in this work are followed. If the decision-maker chose to apply complete preference towards MHSI over a 24-month period, Orleans Parish avoids $1.58 million but achieves a greater decrease in the severity of cases. Alternatively, complete preference toward ELI avoids $66 million with no emphasis on treating the more severe cases. Ultimately, Orleans Parish could expect to see an increase of 7,973 healthy individuals when considering each objective equally, over a do-nothing approach.

### 4.4 Spatiotemporal Visualization

The resource assignment framework also provides decision-makers the ability to visualize clinician allocation on a spatiotemporal scale, based on their recovery preferences. Figures 7 and 8 show this optimal allocation of social workers and psychologists, respectively. In this case study, MHSI and ELI were given equal weights, which results in the allocation of 891 social workers to 'hotspots' of severe mental health cases and high economic loss at the six-month time step (Fig. 7). As treatment proceeds over the next 18 months, social workers begin to spread across more census tracts, as census tract-level concentrations of cases fall. The same interpretation can be made of Figure 8, though dispersion between time steps is less pronounced as there are fewer psychologists (52) and a smaller number of severe cases, relative to mild and moderate. Nonetheless, psychologist dispersion occurs as severe hotspots are reduced by 24 months.

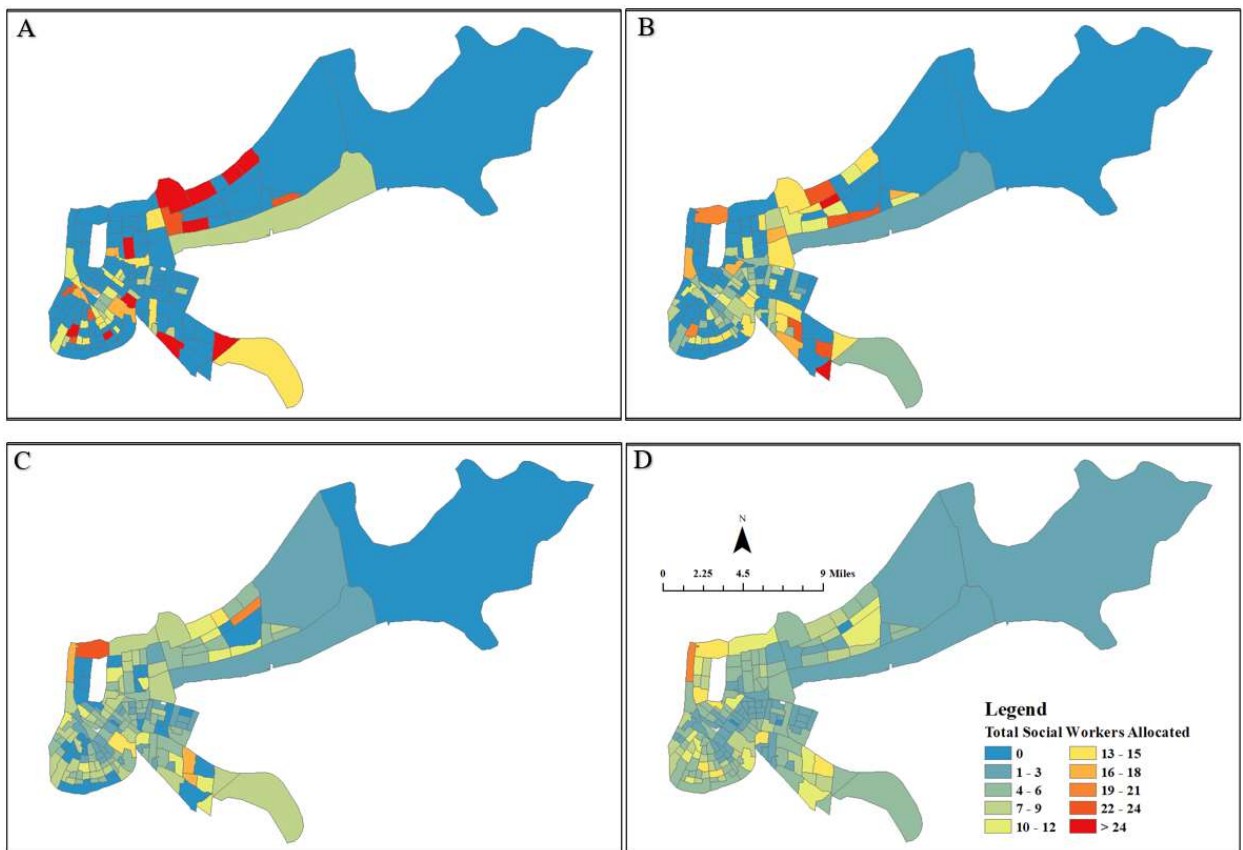

**Figure 7: Orleans Parish Social Worker Allocation with equal preference given to MHSI and ELI; A) 6-12 months, B) 12-18 months, C) 18-24 months, and D) 24-30 months.**

580

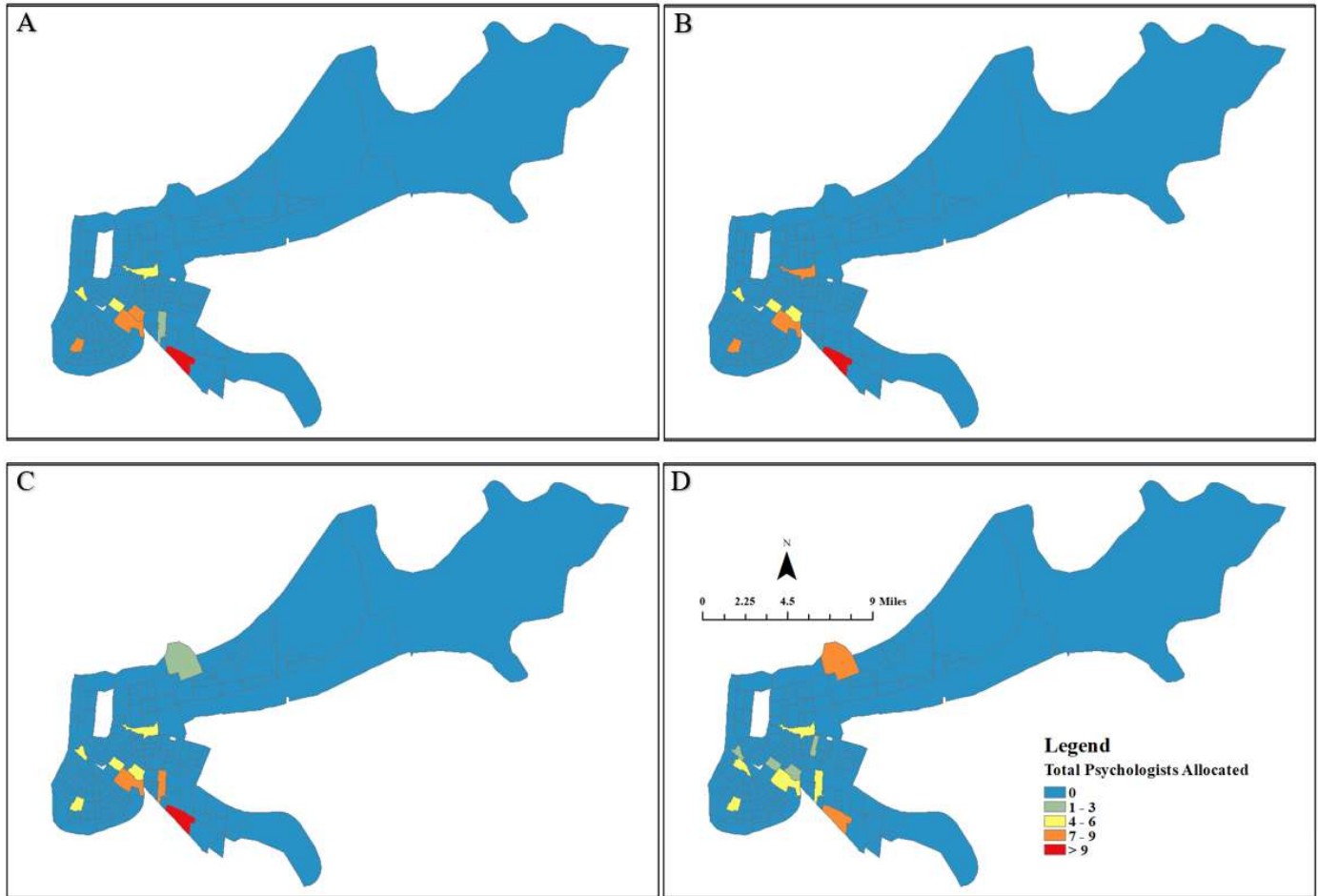

**Figure 8: Orleans Parish Psychologist Allocation with equal preference given to MHSI and ELI; A) 6-12 months, B) 12-18 months, C) 18-24 months, and D) 24-30 months.**

**5 Discussion**

585     The literature concludes that communities can identify areas of social vulnerability and how a disaster, such as a hurricane, can cause mental health impacts across all segments of an exposed population. However, there lacked a clear link between affected communities and how resources should be allocated to address this vulnerability. Furthermore, there was no clear direction as to what aspects of community-scale vulnerability decision-makers should consider when making mental health resource allocation decisions.

590

    The results of this simulation-optimization research show that it is possible to link social vulnerability with psychological impacts of disasters, and that through weighing tradeoffs in treatment options, decision-makers should be able to make efficient and informed resource allocation decisions. Applying the SVI as an operationalized measure of social vulnerability provides decision-makers the capability to weigh treatment tradeoffs to make efficient and informed resource allocation decisions. Given that the

595     SVI is a composite index of socio-economic indicators, decision-makers can distill the SVI's sub-components to tailor their mental health disaster response based upon specific mental health illnesses, patient vulnerability, and the experienced hazard. However, it is important to consider that social vulnerability describes complex relationships between demographic factors and that one factor alone may not necessarily cause that individual to be more vulnerable than another (Flanagan et al. 2011). Rather, it is the

interactions between these factors that provide insight into a population's vulnerability to disaster-induced mental health illness (Cutter and Morath 2013). Furthermore, vulnerability can be introduced by the decision-makers themselves if their disaster planning and subsequent response fails to meet the needs of all populations within the community (Flanagan et al. 2011). However, decision-makers have the opportunity to account for the pitfalls of considering all or partial components of the SVI through coupled simulation-optimization outcomes. The resource assignment framework allows decision-makers to influence the optimization based on their preference and community needs in its current configuration. This framework also satisfies a need as defined by the literature where state and local agencies may need a system to allocate the resources necessary post-disaster (Flanagan et al., 2011).

The resource assignment framework's value lies in the efficient allocation of resources, though the results presented here are a limited case study. Through user definition, the resource assignment framework produces vastly different decision spaces, depending on many factors, including the type and number of resources made available during each treatment cycle, the number of treatment cycles, and definition of decision-maker objectives. Through prioritization of MHSI and ELI, the decision-maker affects when and where resources will be applied. To that end, post-disaster literature has argued that an abundance of resources is made available for disaster recovery and that historically, those resources have been underutilized or mismanaged due to lack of a robust distribution framework ("Hurricane Katrina: A Nation Still Unprepared" 2006). The resource assignment framework provides decision-makers with a mechanism to allocate resources with limited waste. Louisiana and Mississippi, who supplemented their own emergency resources with those of other states in response to Hurricane Katrina and formed an Emergency Management Assistance Compact (EMAC), are an example of how communities may have greater resources available to them when conducting emergency response ("Hurricane Katrina: A Nation Still Unprepared" 2006).

Alternatively, a community may also have limited resources within which they can allocate towards recovery. As discussed in the introduction, communities struggled to obtain mental health resources for students after Hurricane Michael (Jordan 2019). The resource assignment framework provides a case in which limited resources can be utilized most effectively, and the case study provided in this research is closer to resource-limited than it is resource-abundant.

Though the proposed simulation-optimization framework was only utilized for a case study involving one disaster, its iterative approach provides the opportunity to account for serial multi-hazard events as well. Given the potential for prolonged disaster-induced mental health illness, individuals in the midst of recovery from one disaster may experience another disaster. The framework proposed in this paper provides an avenue to assess the cumulative effects of multi-hazard exposure on mental health. The multi-hazard use case provides additional support for the need to minimize the mental health illness outcomes post-disaster and facilitate rapid recovery prior to the next disaster.

Finally, although the case study analyzed within this paper allocated resources to the census-tract level, findings from Hurricane Michael suggest this spatial scale may not be granular enough depending on community needs. For example, it is sometimes necessary to be more specific in where mental health clinicians are deployed such as assigning them to schools where children have easier access to recovery resources. Even so, the resource assignment framework is capable of handling different and varied resource endowments, and it can be calibrated to any spatial scale for which data is provided.

**6 Limitations**

It is imperative to identify the current limitations of this research to provide the appropriate context for the exposition of results. Four primary limitations and one assumption exist that provide opportunities for future work to improve upon.

The first limitation is the use of uniform distributions. All state transition probabilities are applied uniformly across Orleans Parish, meaning that each individual in the parish has equal chances of becoming ill and recovering. This approach functions as a proof of concept to show how a generic, spatially homogenous storm might impact Orleans Parish. This allows the research to focus on a holistic resource allocation and recovery model, as the nature of the event perturbation is less important than using the results to establish a robust framework. Clearly, this model's spatial nature can utilize specific perturbances as an input to the coupled-state

transition and optimization models. However, improvements may be seen by varying state transition probabilities due to individuals' proximity to or damage caused by the storm. This could also be accomplished through varying the probability distributions by population type (i.e., children, adults, elderly, etc.), which would require additional data to determine the relationships between proximity or damage and the population type.

The second limitation is that the optimization function does not permit clinicians to travel between tracts. As such, clinicians only treat patients within the census tract in which they are assigned. This limitation is not addressed in this iteration of the framework as the treatment capacity of each clinician represents the maximum number of patients they could treat without negative impacts to treatment quality. Allowing for residual treatment capacity reduces the likelihood of clinician burnout or degraded treatment quality. However, the decision-maker may choose to address cases of excess clinician demand and capacity in neighboring tracts

exist. An optimized, nearest neighbor framework could be implemented to allow for tract-to-tract travel of clinicians to reduce residual patient demand when clinician capacity surplus exists. A simple, unoptimized analysis of residual capacity and demand within adjacent census tracts illustrates the potential implementation of a second round of optimization at each time step (Fig. 9). In this case, clinicians utilize their excess capacity to reduce the neighboring census tracts' highest residual demand.

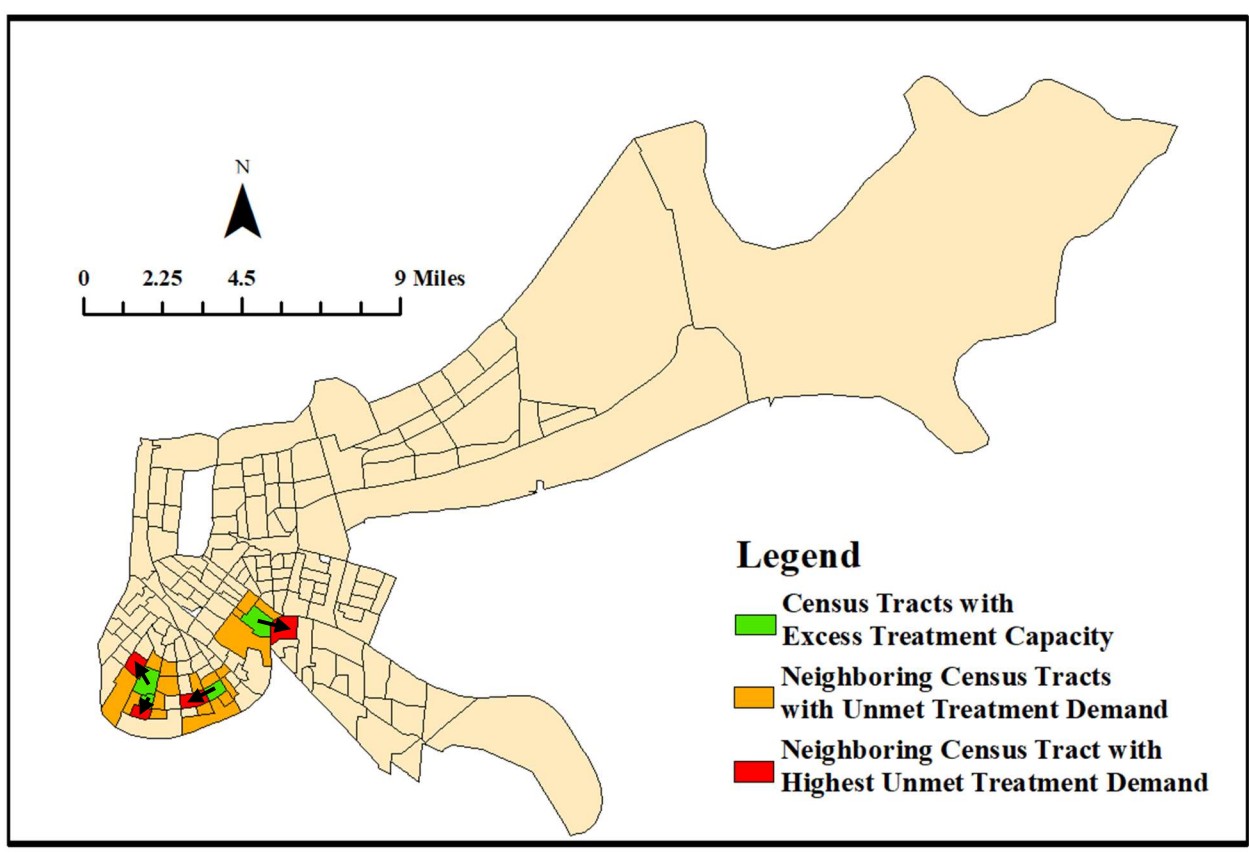


**Figure 9: Theoretical, unoptimized approach to address neighboring treatment limitation for Social Workers 12 months post-hurricane. Arrows show the directional flow of excess treatment capacity.**

The third limitation is in the case study's application of clinician capacity. The available pool of clinicians is likely an overestimate
as local-level resources are more likely to be drawn upon in response to disaster-induced mental health illnesses. State-wide registered clinicians were utilized in lieu of New Orleans-specific clinician data, which was difficult to obtain. However, the model could be easily modified to account for more representative clinician capacity if that data is known.

The fourth limitation is in the model's consideration of input parameters, objectives, and constraints. The current instantiation
suffices as a proof-of-concept; however, future iterations of the model must address realistic factors. For example, considering all state-registered psychologists and social workers as available to provide a generalized treatment with a fixed probability of success does not provide a meaningful decision aid. The parameters do not approach a true representation of reality and would thus present decision-makers an inaccurate estimation of community recovery. With that said, the simulation-optimization model provides the framework necessary to produce a more accurate decision aid. All that is required are inputs that approach the true representation
of reality. Mental health disaster response protocols provide a wealth of possible treatments in all phases of recovery in addition to strategies improving mental health resilience. The model's outputs will converge on accurate representations of reality by considering all possible treatments in each stage of care as well as their efficacy in illness recovery. However, it is worth noting that as the complexity in the spatial-temporal relationships between treatments and resiliency is accounted for, the resulting model will also become increasingly complex. Future researchers must balance this complexity to provide the most accurate decision aid
while maintaining usability for the decision-maker, avoiding the tipping point at which the model becomes too complex to be a useful disaster recovery tool.

Finally, cost of treatment is not analyzed within this research as there is an underlying assumption that disaster response is a public service. Therefore, the cost of recovery is traditionally viewed as less important than recovery. This allows those experiencing mental health illness as a result of a disaster to seek treatment free of charge, from the individual's perspective. Ultimately, though cost of recovery is likely to exceed the economic benefits of people returning to work, as measured by the ELI, policy discussion of the tradeoffs between cost and recovery is out of scope for this research.

## 7 Conclusion

Disasters impact more than just physical infrastructure as they can cause negative mental health effects based on the event's duration, severity, and proximity. The coupled-state transition and optimization framework developed here provides a method that enables communities to overcome the difficulties associated with post-event planning, especially with constrained resources. Through optimization, the allocation of mental health recovery resources is achieved based on balancing preferences in treating the most severe cases as well as economic recovery. A spatial and temporal distribution visualization was used to visualize how the allocation of mental health resources change over time to provide emergency planners a broader context of the optimization results. Though the case study analyzed within this research was specific to a hurricane and fixed resource levels, the resource assignment framework is flexible in many ways due to its novelty as a simulation-optimization framework. This flexibility is seen in the framework's decision criteria, resource optimization, and event perturbance.

First, the resource assignment framework is flexible in the decision criteria under which it optimizes clinician allocation. The current iteration of the resource assignment framework considers ELI and MHSI, but it can be expanded to account for additional indices based on stakeholder needs. For example, a cost of treatment index could be included to add tradeoff consideration for the cost of deploying mental health clinicians for each treatment cycle.

Second, the flexibility in the resource assignment framework is also seen in the context of resource allocation optimization. This research considers two human resources: social workers and psychologists. In terms of mental health recovery, the resource assignment framework-considered resources could also take the form of hospital beds or medication, as an example. Considering emergency response more broadly, the resource assignment framework can be utilized in various applications from human resource allocation, as discussed in this research, to physical resource distribution. Each of these applications could help inform policy and operational decisions based on community needs in post-disaster environments.

Ultimately, it is important to remember the resource assignment framework is not event-specific. The authors recognize that events have spatial patterns, but rather than creating a model with limited spatial context, a uniform approach was used to stress the resource assignment framework and provide meaningful results. Establishing a case study considering a storm with uniform impacts across a spatial scale represents a conservative approach that is designed to stress the model spatially and ensure the resource assignment framework is allocating resources in an expected manner. With this accomplished, further research can now investigate events such as a bomb blast, where the decision-maker might expect clinician allocation to take the form of concentric rings around the blast site, or a tornado that may have a more linear allocation compared to a hurricane, which is more representative of a uniform allocation. In respect to a bomb blast event, the resource assignment framework can utilize recent research evaluating

probability of facility destruction, as well as the facility damage level, subsequent personnel loss, and psychological effects resulting from the blast to inform optimal resource allocation during event recovery (Schuldt and El-Rayes 2018; Schuldt et al. 2019). With many ways to advance this research, the resource assignment framework provides the first steps toward informed decision making in terms of constrained resource optimization in response to disasters.

**Notation**

**Table 2: Optimization Model Variable Definitions**

| Variable | Definition |
|---|---|
| $MHS_{NT}$ | Mental Health Severity given no treatment |
| SVI | Social Vulnerability Index |
| $w_M$ | Mild case weight factor for severe equivalence |
| M | Number of mild mental health illnesses |
| $w_{Mod}$ | Moderate case weight factor for severe equivalence |
| Mod | Number of moderate mental health illnesses |
| S | Number of severe mental health illnesses |
| $MHS_T$ | Mental Health Severity given treatment |
| UD | Remaining demand after clinician allocation |
| $P_{Cap}$ | Number of patients psychologists can treat |
| P | Number of psychologists allocated |
| $SW_{Cap}$ | Number of patients social workers can treat |
| $SW_M$ | Number of social workers allocated to treat mild cases |
| $SW_{Mod}$ | Number of social workers allocated to treat moderate cases |
| MHSI | Mental Health Severity Index |
| $EL_{NT}$ | Economic Loss given no treatment ($) |
| $EL_T$ | Economic Loss given treatment ($) |
| $EL_M$ | Economic Loss of mild cases measured in daily productivity loss ($) |
| $EL_{Mod}$ | Economic Loss of moderate cases measured in daily productivity loss ($) |
| $EL_S$ | Economic Loss of severe cases measured in daily productivity loss ($) |
| ELI | Economic Loss Index |
| $w_1$ | Objective function weight factor for MHSI |
| $w_2$ | Objective function weight factor for ELI |
| MI | Optimization objective to minimize mental health effects of disasters |

**Code and Data Availability**

The code developed during this study is available from the corresponding author upon request. The data analyzed during this study is publicly accessible from https://www.atsdr.cdc.gov/placeandhealth/svi/data_documentation_download.html. The data includes 2016 census tract data from Louisiana.

**Author Contribution**

Stephen Cunningham developed the model code and prepared the manuscript. Steven Schuldt developed the methodology for the optimization model and reviewed the final manuscript. Christopher Chini reviewed both the visualization of the data and the final manuscript. Justin Delorit developed the methodology for the simulation model, provided supervision of research activities, and reviewed the final manuscript.

**Competing Interests**

The authors declare that they have no conflict of interest.

**Disclaimer**

The views expressed in this paper are solely those of the authors and are not reflective of the official policy or position of the United States Air Force, Department of Defense, or the United States government.

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
