# Peer review of "A simulation-optimization framework for post-disaster allocation of mental health resources"

_Natural Hazards and Earth System Sciences, 2020_

## Author Comment (AC1)

We are thankful to the Referee for providing meaningful and constructive feedback. We are hopeful that the amendments we offer in this round improve the quality and clarity of the manuscript.

**Comment:** Line 35 onwards: Do you have any information for comparison - years before the event or the situation one year later?

**Reply:** Yes, we agree that adding comparison information will benefit this section. Little could be found regarding follow on studies beyond the one-year mark after Hurricane Michael's landfall. However, data one year after the storm as well as data from previous hurricanes can provide context to the long-term effects of the hurricane on mental health. The following will be added:

Additionally, one year after Hurricane Michael's landfall, one-third of the affected population, in both Bay and Gulf counties, is expected to have worsening anxiety, depression, or insomnia (Rodriguez et al. 2021). In attempt to estimate the long-term effects of the hurricane, historical context can be applied. For example, 20 months after Category 4 Hurricane Irma made landfall in Florida, 17% of those in the storm's path reported being anxious while 11.3% reported signs of depression (Torres-Mendoza et al. 2021). Due to the continued prevalence of disaster-induced mental health illnesses, Torres-Mendoza et al. 2021 recommend that emergency preparedness plans emphasize mental health services especially in the context of long-term recovery.

**Comment:** Section 2.1: I recommend including further recent reviews and meta-analysis studies in this context to provide a wider overview and more details in your study. E.g.: to name only a few important articles:

**Reply:** We agree that additional detail can be added to Section 2.1 to further explain the relationship between disasters and mental health. The following will be added to section 2.1:

The relationship between disasters and mental health is further analyzed through a series of systematic reviews and meta-analyses. In a 6-year, longitudinal study of PTSD after the Indian Ocean earthquake and tsunami, the onset of PTSD was found to be one month post-disaster while the majority of those impacted recovered within three years (Arnberg, Johannesson, and Michel 2013). Interestingly, higher rates of depression and alcohol abuse were not associated with natural disaster exposure as in other studies; however, Arnberg et al., 2013 still bring attention to the persistent nature of disaster-induced negative mental health impacts.

In attempt to determine the relationship between exposed and non-disaster exposed individuals, a meta-analysis was conducted to compare psychological distress and psychiatric disorder rates post-disaster. Compared with non-exposed populations, those with exposure experienced a higher degree of psychological distress, as much as 1.84 times that of those with no exposure to a natural disaster (Beaglehole et al. 2018). Additionally, some experience even higher rates of mental health illness than others. For example, older

adults are 2.11 times more likely to experience PTSD and 1.73 times more likely to develop an adjustment disorder (Parker et al. 2016). This is consistent with other findings in the literature review that indicate the elderly are at risk in terms of disaster-induced mental health illness (Ursano et al., 2003; USGCRP 2016).

Exploring the efficacy of medical interventions to treat those affected by a disaster are imperative in rehabilitation activities. This efficacy can be used to help determine which treatments are best suited for disaster-response activities. Medical interventions ranged from community-based psychosocial programs, Neuro Emotional Technique (NET), school-based intervention, and social group work (Khan et al. 2015). One study did not show a significant improvement in mental health with the introduction of Institution-based rehabilitation therapy for earthquake survivors; however, the remaining studies did show significant improvement in mental health outcomes due to the medical intervention (Khan et al. 2015). Specifically, Beger and Gelkopf found that 82% of probably PTSD cases improved when a school-based intervention was used to reduce stress-related symptoms of Tsunami exposure (Khan et al. 2015).

**Comment:** Section 2.2: please provide also scientific literature in this context. What is the difference to SoVI developed from Susan Cutter and her team and widely used in the U.S.? Why did you not apply this scientifically proofed approach? see: http://artsandsciences.sc.edu/geog/hvri/publications

**Reply:** We agree that a comparison between Cutter's SoVI and the CDC's SVI is needed to provide a clearer picture of the current state of the field. Below is our addition to Section 2.2.

A second method of measuring social vulnerability aims to provide more context to nation-level vulnerability risk by accounting for social attributes such as socio-economic status, race, gender, age, employment, and housing considerations (Cutter and Morath 2013). Cutter and Morath 2013's Social Vulnerability Index (SoVI) uses United States Census variables, which were down selected via statistical methods designed to remove multicollinearity. Both Cutter and Flanagan et al., 2011 emphasize that social vulnerability is not derived from one demographic, but from interactions among demographic categories Additionally, both authors agree that the level of analysis must be geographically granular enough to distinguish demographical differences. The primary difference between the two methods is that the CDC's SVI can be operationalized as a toolkit, capable of informing decision-makers with easily accessible and understandable data. While SoVI's underlying algorithms have been updated, the SVI improves based on user feedback. User feedback is an integral component of product design. The SVI becomes more useful as decision-makers see how the toolkit adapts to their needs and provides actionable data.

**Comment:** Section 2.3.: Please provide a short input about other possible models in this context.

**Reply:** We agree that additional input into other treatment methods is required. The following will be added to Section 2.3.

Other possible models in addition to traditional treatment through primary health-care providers includes community-based programs and task shifting (Kakuma et al. 2011). Task shifting aims to provide some level of care to those without access to specialists. Clinicians with fewer qualifications will receive more specific training to account for the needs of the at-risk populations (Javadi et al. 2017). The case study we explore in this research focuses on providing treatment through psychologists and social workers. Given that these resource pools are limited, incorporating community-based programs and task shifting could expand the pool of available resources to better aid the affected population in their recovery.

**Comment:** Section 2.4.: Please check if you can find more recent estimations for this part or provide a methods to adapt this to the current situation (indexation of the costs, ...)

**Reply:** Yes, we were able to find more information that can be applied to the context cost estimates, but also provide insight into how the model parameters can be changed to account for changing wage loss estimations. The following is our response:

In addition to individualized economic loss, poor mental health can negatively impact economic growth via direct and indirect costs where direct costs include the treatment of the illness while indirect costs include income loss (Trautmann, Rehm, and Wittchen 2016). Between 2011-2030, cumulative economic output loss due to mental health illness is projected to total \$16.3 trillion globally (Trautmann, Rehm, and Wittchen 2016).

While this study does not provide a direct estimation for an individual's average indirect cost due to mental health, it provides support for the coupling of Birnbaum et al's estimation of hours lost and Zahran et al's wage loss estimate due to poor mental health to provide an economic loss metric that informs optimal mental health clinician allocation for wage restoration purposes). As research advances, the methodology in accounting for economic loss due to mental health illnesses can be modified through changing the inputs to the $EL_M$, $EL_{Mod}$, and $EL_S$ variables. This will account for both wage changes and changes in the estimation of how many days away from work an individual will experience.

**Comment:** Line 498: please provide citations. Do which studies to you refer to?

**Reply:** This is our synthesis of the information gathered in the literature review rather than a new idea with a direct citation.

**Citations**

Arnberg, Filip, Kerstin Johannesson, and Per-Olof Michel. 2013. "Prevalence and Duration of PTSD in Survivors 6 Years after a Natural Disaster." Journal of Anxiety Disorders 27 (3): 347–52. https://doi.org/10.1016/j.janxdis.2013.03.011.

Beaglehole, Ben, Roger Mulder, Chris Frampton, Joseph Boden, Giles Newton-Howes, and Caroline Bell. 2018. "Psychological Distress and Psychiatric Disorder after Natural Disasters: Systematic Review and Meta-Analysis." The British Journal of Psychiatry 213 (6). https://doi.org/10.1192/bjp.2018.210.

Cutter, Susan, and Daniel Morath. 2013. "The Evolution of the Social Vulnerability Index
    (SoVI)." United Nations University Press.
    http://collections.unu.edu/eserv/UNU:2880/n9789280812022_text.pdf.

Parker, Georgina, David Lie, Dan Siskind, Melinda Martin-Khan, and Beverly Raphael.
    2016. "Mental Health Implications for Older Adults after Natural Disasters - a
    Systematic Review and Meta-Analysis." International Psychogeriatrics 28 (1): 11–
    20. https://doi.org/10.1017/S1041610215001210.

Khan, Fary, Bhasker Amatya, James Gosney, Farooq Rathore, and Frederick Burkle Jr.
    2015. "Medical Rehabilitation in Natural Disasters: A Review." Archives of Physical
    Medicine and Rehabilitation 96 (9): 1709 1727.
    https://doi.org/10.1016/j.apmr.2015.02.007.

Kakuma, Ritsuko, Harry Minas, Nadja Ginneken, Mario Dal Poz, Keshav Desiraju, Jodi
    Morris, Shekhar Saxena, and Richard Scheffler. 2011. "Human Resources for Mental
    Health Care: Current Situation and Strategies for Action." The Lancet 378 (9803):
    1654–63. https://doi.org/10.1016/S0140-6736(11)61093-3.

Javadi, D, I Feldhaus, A Mancuso, and A Ghaffar. 2017. "Applying Systems Thinking to
    Task Shifting for Mental Health Using Lay Providers: A Review of the Evidence."
    Global Mental Health 4 (14). https://dx.doi.org/10.1017%2Fgmh.2017.15.

Trautmann, Sebastian, Jurgen Rehm, and Hans-Ulrich Wittchen. 2016. "The Economic
    Costs of Mental Disorders." Embo Reports 17 (9): 1245–49.
    https://dx.doi.org/10.15252%2Fembr.201642951.

Torres-Mendoza, Yaritbel, Alison Kerr, Amy Schnall, Carina Blackmore, and Summer
    Hartley. 2021. "Community Assessment for Mental and Physical Health Effects
    After Hurricane Irma - Florida Keys" 70 (26): 937–41.
    http://dx.doi.org/10.15585/mmwr.mm7026a1.

Rodriguez, Edda, Chris Duclos, Jessica Joiner, Melissa Jordan, Keshia Reid, and Kristina
    Kintziger. 2021. "Community Assessment for Public Health Emergency Response
    (CASPER) Following Hurricane Michael, Bay and Gulf Counties, Florida, 2019."
    Journal of Public Health Management and Practice, June.
    https://doi.org/10.1097/PHH.0000000000001365.

---

## Author Comment (AC2)

We are thankful to the Referee for providing meaningful and constructive feedback. We are hopeful that the amendments we offer in this round improve the quality and clarity of the manuscript.

**Comment:** 1. Although the authors review literature well in Section 2, it is difficult to know the current methods and protocols for treating disaster-related mental health problems. For example, what are the current standards and guidelines for allocating mental health clinicians and other resources following a disaster event? I expected to see this in Section 2.3, but this section describes methods for analyzing treatment measures rather than treatment options.

**Reply:** We agree that there could be more discussion involving current methods and protocols for post-disaster mental health treatment. The following will be added to section 2.3 at line 138 to add more clarity to this point.

Mental health care post-disaster is generally organized into three phases: early, intermediate, and long-term interventions (Hierholzer, Bellamy, and Mannix 2015). Early interventions range from Cognitive Behavioral Therapy to Psychological First Aid; intermediate interventions range from Classroom-Based Intervention to Specialized Crisis Counseling; while long-term interventions range from Cognitive Processing Therapy to Systematic Desensitization (Hierholzer, Bellamy, and Mannix 2015). Typically, these interventions will be performed by clinicians within medical facilities and shelters, while also facilitating community-based recovery to improve upon mental health resiliency in the event of future disasters ("Disaster Behavioral Health" 2020).

Cohen 2002 describes a similar approach in which treatment is distributed throughout three phases: impact, short-term, and long-term. Though different in name, these phases align similarly to those proposed by Hierholzer et al., 2015. However, Cohen 2002 introduces a new element in which treatment can target five levels of disaster-impacted individuals. Behavioral health needs post-disaster ranges from level one, primary survivors, to fifth-level victims. Primary survivors are those who experienced the disaster first-hand while fifth-level victims are those who experience some form of distress after learning about the event (Cohen 2002). Each of the five levels will have varying recovery needs that will impact clinician allocation. While it is important to consider the three phases of treatment post-disaster, it is also imperative to introduce preventative medicine as an opportunity to decrease the mental health impact of a disaster (Math et al. 2015). Preventative medicine manifests itself in terms of readiness in which training and equipping communities with mental health recovery tools prior to a disaster can improve the resilience of disaster-impacted individuals.

**Comment:** 2. The SVI is a composite index that combines multiple social and economic indicators to represent overall vulnerability. However, as the authors explain in Section 2.1, a tailored vulnerability based on a set of specific indicators would be more reasonable to model a specific type of mental health illness, patient, and hazard. This could also provide new options for decision-makers. Please elaborate on this point in the proper section.

**Reply:** Thank you for this opportunity to elaborate on this point. We will make these additions to the Discussion after lines 504-506: "The results of this simulation-optimization research show that it is possible to link social vulnerability with psychological impacts of disasters, and that through weighing tradeoffs in treatment options, decision-makers should be able to make efficient and informed resource allocation decisions."

Applying the SVI as an operationalized measure of social vulnerability provides decision-makers the capability to weigh treatment tradeoffs to make efficient and informed resource allocation decisions. Given that the SVI is a composite index of socio-economic indicators, decision-makers can distill the SVI's sub-components to tailor their mental health disaster response based upon specific mental health illnesses, patient vulnerability, and the experienced hazard. However, it is important to consider that social vulnerability describes complex relationships between demographic factors and that one factor alone may not necessarily cause that individual to be more vulnerable than another (Flanagan et al. 2011). Rather, it is the interactions between these factors that provide insight into a population's vulnerability to disaster-induced mental health illness (Cutter and Morath 2013). Furthermore, vulnerability can be introduced by the decision-makers themselves if their disaster planning and subsequent response fails to meet the needs of all populations within the community (Flanagan et al. 2011). However, decision-makers have the opportunity to account for the pitfalls of considering all or partial components of the SVI through coupled simulation-optimization outcomes.

**Comment:** 3. Lines 473-474: I think the baseline model needs to include clinicians to reflect the real world rather than the do-nothing approach. Could you include a baseline model that allocates clinicians according to the number of populations, and then compare it to the optimized results (i.e., Figure 5)? I believe this can better quantify the benefits of this approach.

**Reply:** We agree that the case study applied is an overestimate of clinician capacity as local-level resources are more likely to be drawn upon in response to disaster-induced mental health illnesses. We could not find New Orleans-specific clinician data and opted to use state-wide registered clinicians. However, the model could be easily modified to account for more representative clinician capacity if that data is known. With that said, we will add the overestimate of clinician capacity as a limitation to the study. Currently, the limitations account for a clinician treating patients in neighboring census tracts as well as a clinician's ability to only perform one-on-one treatment sessions. Including estimated clinician capacity will round out these limitations to improve the clarity in the methods upon which we chose to implement the case study.

**Comment:** 4. I recommend explaining additional input parameters, objectives, and constraints in the Limitations section. The current variables are sufficient for a proof-of-concept, but future research must address realistic factors. This could be related to the current standards and guidelines mentioned in my comment #1.

**Reply:** We agree that this is good discussion and will rework the third limitation, one-on-one treatment to address the limitations in model variables in general. Line 559.

The third limitation is in the model's consideration of input parameters, objectives, and constraints. The current instantiation suffices as a proof-of-concept; however, future iterations of the model must address realistic factors. For example, considering all state-registered psychologists and social workers as available to provide a generalized treatment with a fixed probability of success does not provide a meaningful decision aid. The parameters do not approach a true representation of reality and would thus present decision-makers an inaccurate estimation of community recovery. With that said, the simulation-optimization model provides the framework necessary to produce a more accurate decision aid. All that is required are inputs that approach the true representation of reality. Mental health disaster response protocols provide a wealth of possible treatments in all phases of recovery in addition to strategies improving mental health resilience. The model's outputs will converge on accurate representations of reality by considering all possible treatments in each stage of care as well as their efficacy in illness recovery. However, it is worth noting that as the complexity in the spatial-temporal relationships between treatments and resiliency is accounted for, the resulting model will also become increasingly complex. Future researchers must balance this complexity to provide the most accurate decision aid while maintaining usability for the decision-maker, avoiding the tipping point at which the model becomes too complex to be a useful disaster recovery tool.

**Comment:** Lines 66-67: "before constructing a model" is repeated.

**Reply:** Thank you for catching the duplication of this phrase. This will be fixed in the next revision.

**Comment:** Lines 379-381: Could you include population and SVI maps in the manuscript or supplement file? This will assist readers in identifying the optimal allocations spatially across socio-economic statuses.

**Reply:** Yes, we concur that this would ground the readers in an understanding of the population and vulnerability distributions across the area of study. This map will be added to line 381 which will provide an accompanying visualization for the statement of population and SVI minimum and maximum values.

[Figure]

Figure 4: A) New Orleans population by census tract. B) New Orleans SVI scores by census tract.

**Comment**: Lines 230-245: Please provide references to support these sentences.

**Reply:** The content from these lines is original to the authors and describes the phased approach to the coupled simulation-optimization model as well as an example of how phase 1 can vary in implementation.

**Comment:** Equations (2-3) and (6-7): Please specify the subscripts "NT" and "T".

**Reply:** We agree that the current text lacks specification that those subscripts mean No Treatment (NT) and Treatment (T). Line 333-334 will now read: "Equation 2 provides the calculation for a census tract's MHS, given that no clinicians are allocated to conduct treatment (MHS$_{NT}$). Line 336 will now read: "Equation 3 provides a measure of how MHS improves with clinician allocation (MHS$_T$).

**Comment:** Equations (5 and 8): Please specify the subscript "j" and use "N" or "J" for all census tracts.

**Reply:** We agree that the variable descriptions are confusing. The description of $j$ is in line 301 and we will make the edit to Line 349: "This becomes an index [0,1], where a value of zero represents a complete reduction of MHS across all census tracts (J), and a value of one indicates no improvement."

Equation 5 now reads: $MHSI = \dfrac{\Sigma_{j=1}^{J} MHS_{T,j}}{\Sigma_{j=1}^{J} MH\ _{NT,j}}$

Equation 8 now reads: $ELI = \dfrac{\Sigma_{j=1}^{J} EL_{T,j}}{\Sigma_{j=1}^{J} EL_{NT,j}}$

**Comment:** Please explain a potential application of agent-based modeling for clinician allocation, as a recent study does (Lines 205-206).

**Reply:** We agree that additional discussion of agent-based modeling provides a useful avenue for future work to explore. The following is additional discussion after Line 206:

Sun and Zhanmin 2020 describe that agent-based modeling can be used in the context of reinforcement learning where the agent's decision results in some reward which represents system improvement. Agent-based modeling provides an alternative approach to the simulation-optimization methodology proposed in this paper. In an agent-based modeling approach, clinicians would have the ability to choose which treatment to provide each patient, while the reward is the improvement, or degradation, of their mental health. The overall state of the system is then the quality of the community's mental health. The goal would remain the same: minimizing the mental health impact of the disaster. However, agent-based modeling would allow the clinicians to make treatment decisions and learn from the resulting mental health outcomes of those choices. This method could be increasingly useful as the complexity of the model also increases with the addition of more treatment options and treatment efficacy.

**Comment:** Given the frequency of natural disasters (e.g., hurricanes), the proposed approach can be applied to serial multi-hazard events. I recommend expanding on this benefit as well.

**Reply:** We agree that this is a good point to address in the discussion section. The following will be added to the discussion:

Though the proposed simulation-optimization framework was only utilized for a case study involving one disaster, its iterative approach provides the opportunity to account for serial multi-hazard events as well. Given the potential for prolonged disaster-induced mental health illness, individuals in the midst of recovery from one disaster may experience another disaster. The framework proposed in this paper provides an avenue to assess the cumulative effects of multi-hazard exposure on mental health. The multi-hazard use case provides additional support for the need to minimize the mental health illness outcomes post-disaster and facilitate rapid recovery prior to the next disaster.

Citations:

Hierholzer, Erik, Nikki Bellamy, and Brenda Mannix. 2015. "Disaster Technical Assistance Center Supplemental Research Bulletin: Disaster Behavioral Health Interventions Inventory." SAMHSA, May. https://www.samhsa.gov/sites/default/files/dtac/supplemental-research-bulletin-may-2015-disaster-behavioral-health-interventions.pdf.

"Disaster Behavioral Health." 2020. Public Health and Medical Emergency Support for a Nation Prepared, September. https://www.phe.gov/Preparedness/planning/abc/Pages/disaster-behavioral.aspx.

Cohen, Raquel. 2002. "Mental Health Services for Victims of Disasters." World Psychiatry 1 (3): 149–52.

Math, Suresh, Maria Nirmala, Sydney Moirangthem, and Naveen Kumar. 2015. "Disaster Management: Mental Health Perspective." Indian Journal of Psychological Medicine 37 (3): 261–71. https://dx.doi.org/10.4103%2F0253-7176.162915.

Cutter, Susan, and Daniel Morath. 2013. "The Evolution of the Social Vulnerability Index
(SoVI)." United Nations University Press.
http://collections.unu.edu/eserv/UNU:2880/n9789280812022_text.pdf.